# DUMoE: Deep Unfolding Mixture-of-Experts for Compressive Imaging

## Abstract

Deep Unfolding-based Networks (DUNs) have attracted attention due to their high performance and a certain degree of interpretability. However, existing DUNs often lack flexibility in handling details and features in different images during reconstruction, as they typically involve multiple iterative modules cascading through the same structure for each iteration. To address this limitation, we propose DUMoE, a novel sparsely-activated Deep Unfolding Mixture-of-Experts (MoE) architecture for Compressive Imaging (CI). By integrating the deep unfolding paradigm into the MoE, we enable DUMoE to adaptively reconstruct various images by utilizing different experts at each iteration stage. Specifically, we unfold traditional SpaRSA iterations into experts within DUMoE and employ top-1 switch routing to save computational consumption and enhance flexibility. Additionally, we introduce the Degradation-Aware Mask within the self-attention mechanism to prioritize image degradation caused by dimensionality reduction in CI, thereby enhancing reconstruction fidelity. Moreover, we incorporate the Multi-Scale Gate to improve the DUMoE's adaptability to image features at different scales and facilitate information transmission across iteration stages. Extensive experiments across various CI recovery tasks, including natural image compressive sensing, magnetic resonance imaging, and snapshot compressive imaging, demonstrate the superior performance and effectiveness of DUMoE. To the best of our knowledge, we are the first to leverage the deep unfolding paradigm within the MoE framework.

## 1 Introduction

Compressive Imaging (CI) is an imaging methodology that leverages signal sparsity or compressibility principles to enable high-fidelity image reconstruction using markedly fewer samples than conventional methods (Candès & Wakin (2008)). This capability allows CI to dramatically reduce sampling complexity and data storage requirements, while concurrently enhancing imaging speed and efficacy. Therefore, CI finds extensive applications across diverse domains, particularly in natural Image Compressive Sensing (ICS) (Kulkarni & Turaga (2015); Zhang & Ghanem (2018); Zha et al. (2023)), CS Magnetic Resonance Imaging (CS-MRI) (Lustig et al. (2007; 2008); Yang et al. (2016)), and Snapshot Compressive Imaging (SCI) (Ma et al. (2019); Yuan et al. (2021); Cheng et al. (2023)). Specifically, assuming that $\mathbf{x} \in \mathbb{R}^N$ denotes the vector of representation coefficients of original signal, $\mathbf{A} \in \mathbb{R}^{M \times N}(M \ll N)$ denotes the linear sampling matrix, and $\mathbf{y} \in \mathbb{R}^M$ is the measurement obtained from underdetermined system $\mathbf{y} = \mathbf{A}\mathbf{x}$, traditional CI recovery problem can be formulated as follows:

$$\min_{\mathbf{x}} \frac{1}{2}\|\mathbf{A}\mathbf{x} - \mathbf{y}\|_{\ell_2}^2 + \tau \mathcal{R}(\mathbf{x}), \tag{1}$$

Existing methods for CI recovery problem can be classified into three categories: traditional optimization methods, purely Deep Learning (DL)-based networks, and Deep Unfolding-based Networks (DUNs). First, traditional optimization methods, such as Iterative Shrinkage/Thresholding Algorithm (ISTA) (Beck & Teboulle (2009)), Alternating Direction Method of Multipliers (ADMM) (Boyd (2010)), Fixed-Point Continuation (FPC) (Hale et al. (2008)), Sparse Reconstruction by Separable Approximation (SpaRSA) (Wright et al. (2009)), and others (Bioucas-Dias & Figueiredo (2007); Goldstein & Osher (2009); Chambolle & Pock (2011); Donoho et al. (2009)), typically rely on iterative steps to gradually optimize results and achieve (sub)optimal outcomes with theoretical guarantees. However, these methods often require hand-crafted parameters fine-tuning, exhibit

limited representation ability across various data, and demand significant computational time to attain satisfactory results. Second, purely DL-based networks (Kulkarni et al. (2016); Gan et al. (2023a;b); Shen et al. (2024)) leverage DL modules such as convolutional neural networks (CNNs), Vision Transformer (ViT) (Dosovitskiy et al. (2021)), and their combinations to learn the mapping relationship between measurements and ground truth images, thus achieving superior reconstruction performance. However, these methods do not possess theoretically proven properties and interpretability, as they often lack insights and knowledge from the CI domain. Third, DUNs (Zhang & Ghanem (2018); Gan et al. (2024b); Yang et al. (2020)) are inspired by traditional iterative optimization algorithms like ISTA, ADMM and FPC. They integrate DL modules into the iterative steps of these optimization algorithms, creating cascaded networks with multiple stages, where each stage represents an iteration within the optimization algorithm. This integration not only leads to fast and accurate CI recovery but also introduces domain-specific prior knowledge inherent to the CI domain.

However, existing DUNs often employ multiple iterative modules cascading through the same module for each iteration, limiting flexibility when handling fine-grained details in diverse images. To overcome this limitation, we propose DUMoE, a sparsely-activated Deep Unfolding Mixture-of-Experts (MoE) framework for CI tasks. Initially, we transform the iterative steps of the SpaRSA algorithm into deep unfolding modules, integrating them as experts within DUMoE. During reconstruction, rather than utilizing all experts, we adopt the top-1 switch routing, thereby reducing computational overhead and enhancing the model's flexibility to handle details and features in distinct images. Additionally, we introduce the Degradation-Aware Mask into the self-attention mechanism to enhance DUMoE's focus on image areas susceptible to degradation in various CI tasks. Furthermore, we integrate U-Block into the Gate modules to leverage multi-scale features for experts selection and enhanced image reconstruction, and improve feature transmission during the reconstruction.

Our contributions can be summarized as follows:

**(i)** We propose DUMoE, a novel sparse MoE framework integrated with deep unfolding SpaRSA. Within DUMoE, we unfold the iterations of traditional SpaRSA into experts, i.e., Deep Unfolding SpaRSA Experts, which are sparsely-activated based on the top-1 switch routing. To the best of our knowledge, we are the first to leverage the deep unfolding paradigm within the MoE framework, yielding state-of-the-art (SOTA) results across various CI tasks.

**(ii)** We introduce the Degradation-Aware Mask within the self-attention mechanism of DUMoE, enhancing its adaptability to image degradation in diverse CI tasks. This refinement allows DUMoE to focus more attentively on degraded image details, resulting in higher-quality reconstructed images.

**(iii)** We incorporate a Multi-Scale Gate into DUMoE, which enhances the capacity of model to capture fine-grained feature across different image scales, and facilitates the transmission of multi-scale features at different stages, leading to significant improvements in reconstruction performance.

Comprehensive comparative analyses between DUMoE and other SOTA methods across ICS, CS-MRI, and SCI, highlight the excellent performance of our proposed DUMoE, demonstrating its effectiveness in various CI tasks.

## 2 RELATED WORKS

In recent, various DUNs have emerged in the fields of ICS, CS-MRI, and SCI, showing significant advancements in image reconstruction. In ICS, researchers have devised DUNs to reconstruct natural images from limited measurements. For instance, Zhang and Ghanem introduced ISTA-Net[+] (Zhang & Ghanem (2018)), which integrates CNNs into ISTA's iterative steps and utilizes them for sparse transform-related proximal mapping. Besides, based on the FPC algorithm, Wang and Gan proposed UFC-Net (Wang & Gan (2024)), which introduces the convolution-guided attention and auxiliary iterative reconstruction block to enhance feature extraction and preservation. Other methods include DPC-DUN (Song et al. (2023b)), NesTD-Net (Gan et al. (2024a)), and LTwIST (Gan et al. (2024b)), among others (Chen & Zhang (2022); Zhang et al. (2020); You et al. (2021); Chen et al. (2022); Mou et al. (2022); Song et al. (2023a); Chen et al. (2023a); Song et al. (2023c); Song & Zhang (2023)). In CS-MRI, methods like ADMM-CSNet (Yang et al. (2020)), HiTDUN (Zhang et al. (2022)), MAPUN (Song et al. (2023a)), along with others (Zhang & Ghanem (2018); Neyra-Nesterenko & Adcock (2022); Gan et al. (2024b;a); Wang & Gan (2024)) have been developed to reconstruct high-quality images from partial Fourier data, enabling faster imaging and reduced data acquisition. ADMM-

Figure 1: The overall structure of DUMoE. DUMoE contains an embedding block, $n$ iteration stages, and a post-block. Here, $\mathbf{y}$ represents the measurement and $\mathbf{x}_f$ denotes the output of DUMoE.

CSNet (Yang et al. (2020)) unfolds and generalizes the ADMM algorithm into a deep architecture, while HiTDUN (Zhang et al. (2022)) facilitates multichannel information transmission between unfolding iterative stages. In the domain of SCI, methods like ADMM-Net (Ma et al. (2019)), DGSMP (Huang et al. (2021)), GAP-Net (Meng et al. (2023)), and others (Cai et al. (2022c); Li et al. (2023b); Dong et al. (2023); Qin et al. (2024); Zhao et al. (2024)), aim to recover 3D hyperspectral images (HSI) from 2D measurements containing spectral channel information. For example, Ma et al. proposed ADMM-Net (Ma et al. (2019)), which transforms the ADMM algorithm into a layerwise structure to learn the sparse representation domain through network training. Besides, Meng et al. introduced GAP-Net (Meng et al. (2023)), which unfolds the generalized alternating projection (GAP) algorithm, utilizing CNNs as denoisers projecting the estimate back to the desired signal space.

Recently, Mixture-of-Experts (MoE) has garnered considerable attention in both Natural Language Processing (Shazeer et al. (2017); Dryden & Hoefler (2022); Fedus et al. (2022); Zoph et al. (2022); Mustafa et al. (2022)) and Computer Vision (Riquelme et al. (2021); Puigcerver et al. (2022); Li et al. (2023a); Chen et al. (2023b); Wang et al. (2023); Ye & Xu (2023)). Typically, an MoE layer comprises many experts sharing the same network architecture, alongside a sparse gating or routing function that directs individual inputs to the top-$K$ experts among all candidates (Shazeer et al. (2017); Fedus et al. (2022)). This approach only requires the computation of $K$ experts for a new input, resulting in fast inference times. For instance, Williams et al. introduced the Switch Transformer (Fedus et al. (2022)), a model with sparsely-activated experts, which replaces the dense feed-forward network (FFN) layer in the Transformer with a sparse Switch FFN layer and enables stability in the training process of large sparse models.

## 3 PROPOSED METHOD

### 3.1 SAMPLING PROCESS

Different CI tasks involve diverse sampling processes. Thus, we offer a broad overview here, with detailed task-specific descriptions in Appendix A.2. Let $\mathcal{F}_{\mathbf{A}}(\cdot)$ denote the sampling function and $\mathbf{x}$ be the original images. The generalized sampling process can be formulated as:

$$\mathbf{y} = \mathcal{F}_{\mathbf{A}}(\mathbf{x}), \tag{2}$$

where $\mathbf{y}$ denotes the obtained measurement derived from $\mathbf{x}$.

### 3.2 RECONSTRUCTION STAGE

As shown in Fig. 1 and Fig. 2, the reconstruction stage includes an embedding module, $n$ iteration stages and a post-block. First, assuming $\widetilde{\mathcal{F}}_{\mathbf{A}}(\cdot)$ represents the initialization function, the process of obtaining an initial estimate $\mathbf{x}^{(0)} \in \mathbb{R}^{C_0 \times H \times W}$ from the measurement $\mathbf{y}$ can be expressed as:

$$\mathbf{x}^{(0)} = \widetilde{\mathcal{F}}_{\mathbf{A}}(\mathbf{y}), \tag{3}$$

where $C_0$ denotes the basic channel count, set to 1 for images in ICS and CS-MRI tasks, and 28 for SCI tasks. The embedding module starts with a $3 \times 3$ convolution to increase the channel count from $C_0$ to $C_1$, followed by a Depth-wise Channel Attention Block (DCAB). The post-block structure mirrors that of the embedding module, albeit in reverse order. Each iteration stage integrates Degradation-Aware Self-Attention, Multi-Scale Gate, and three Deep Unfolding SpaRSA Experts. In the first stage, the channel count is $C_1$, while from the 2-nd to the $(n-1)$-th stage, it is $C_2$, with weights shared across them. Moreover, the channel count of $n$-th iteration stage is $C_3 = C_1 + C_2$.

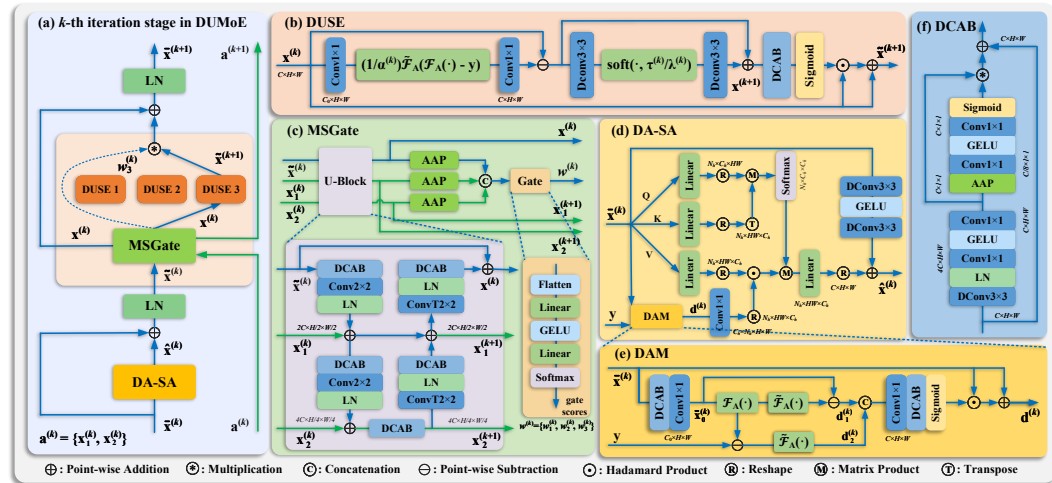

Figure 2: Detailed structure of DUMoE: (a) The $k$-th iteration stage in DUMoE; (b) Deep Unfolding SpaRSA Experts (DUSE); (c) Multi-Scale Gate (MSGate); (d) Degradation-Aware Self-Attention (DA-SA); (e) Degradation-Aware Mask (DAM); (f) Depth-wise Channel Attention Block (DCAB).

### 3.2.1 DEGRADATION-AWARE SELF-ATTENTION

In CI tasks, the reduction in data dimensionality during the sampling process can lead to inevitable loss of information, resulting in image quality degradation characterized by blurring, noise, and distortion. To address this challenge, we introduce a Degradation-Aware Mask (DAM) into self-attention mechanism, proposing Degradation-Aware Self-Attention (DA-SA), as shown in Fig. 2e and Fig. 2d. The DAM incorporates two domains of degradation perception: the image-level domain and the measurement-level domain. Specifically, given $\overline{\mathbf{x}}^{(k)}$ as the input of the DAM in $k$-th iteration stage, $\overline{\mathbf{x}}^{(k)}$ undergoes a DCAB and a $1 \times 1$ convolution to reduce the channel count to $C_0$, yielding the current estimated image $\overline{\mathbf{x}}_0^{(k)}$. On one hand, we quantify the image-level degradation $\mathbf{d}_1^{(k)}$, resulting from reduced-dimensional sampling in CI as follows:

$$\mathbf{d}_1^{(k)} = \overline{\mathbf{x}}_0^{(k)} - \widetilde{\mathcal{F}}_{\mathbf{A}}(\mathcal{F}_{\mathbf{A}}(\overline{\mathbf{x}}_0^{(k)})). \tag{4}$$

On the other hand, we obtain the degradation $\mathbf{d}_2^{(k)}$ in the measurement-level domain by subtracting the initial measurement $\mathbf{y}$ from the measurement of $\overline{\mathbf{x}}_0^{(k)}$, which serves as a data fidelity term to maintain consistency between the measurement of current estimated image and the original measurement:

$$\mathbf{d}_2^{(k)} = \widetilde{\mathcal{F}}_{\mathbf{A}}(\mathbf{y} - \mathcal{F}_{\mathbf{A}}(\overline{\mathbf{x}}_0^{(k)})). \tag{5}$$

Subsequently, we concatenate the $\mathbf{d}_1^{(k)}$ and $\mathbf{d}_2^{(k)}$ along the channel dimension, and use a $1 \times 1$ convolution and DCAB to increase the channel count, which is succeeded by a Sigmoid function to obtain degradation weights for the degraded regions of the images. We then perform a Hadamard product between $\overline{\mathbf{x}}^{(k)}$ and the obtained degradation weights, followed by a residual connection to obtain the output of the DAM, denoted as $\mathbf{d}^{(k)}$. $\mathbf{d}^{(k)}$ then undergoes a $1 \times 1$ convolution to further increase the channel count to $C_h \times N_h$, where $C_h$ denotes the number of channels per head and $N_h$ represents the number of heads. Subsequently, the obtained features are combined with Value $\mathbf{V}$ in DA-SA using a Hadamard product to prioritize attention to the degraded parts and details in the images. The integration of DAM enhances the DUMoE's ability to perceive degraded details, consequently improving feature extraction capabilities and resulting in more representative features.

### 3.2.2 MULTI-SCALE GATE

It is crucial to effectively utilize multi-scale features for recovering fine image details (Mou et al. (2022); Cai et al. (2022b)) in CI tasks. Hence, we introduce a U-Block within the gating module and utilize features at different scales to compute gate scores for expert selection. The structure of Multi-Scale Gate (MSGate) is shown in Fig. 2c. Specifically, we employ a $2 \times 2$ convolution with

stride of 2 to halve the scale of $\widetilde{\mathbf{x}}^{(k)}$ and double the number of channels. After applying residual connections at the same scale, we utilize a $2 \times 2$ transpose convolution to increase the image scale and reduce the number of channels, resulting in three different scales of $\mathbf{x}^{(k)}$, $\mathbf{x}_1^{(k+1)}$, and $\mathbf{x}_2^{(k+1)}$, which are then individually passed through Adaptive Average Pooling (AAP) and concatenated along the channel dimension before being inputted into the Gate. The Gate consists of two linear layers with GELU activation in between, followed by a Softmax operation to obtain corresponding gate scores $w^{(k)} = \{w_1^{(k)}, w_2^{(k)}, w_3^{(k)}\}$. Furthermore, instead of utilizing all three experts, we adopt the top-1 switch routing introduced by Fedus et al. (2022) to sparsify the gating modules and reduce computational overhead. At last, the output is obtained by multiplying the gate score of the corresponding expert with the output of that expert.

### 3.2.3 DEEP UNFOLDING SPARSA EXPERTS

The detailed structure of Deep Unfolding SpaRSA Experts (DUSE) is presented in Fig. 2b. Specifically, we define $f(\mathbf{x}) = \frac{1}{2}\|\mathbf{A}\mathbf{x} - \mathbf{y}\|_{\ell_2}^2$, and we can transfer Eq. (1) into following iterative steps:

$$\mathbf{x}^{(k+1)} \in \arg\min_{\mathbf{z}} (\mathbf{z} - \mathbf{x}^{(k)})\nabla f(\mathbf{x}^{(k)}) + \frac{\lambda^{(k)}}{2}\|\mathbf{z} - \mathbf{x}^{(k)}\|_{\ell_2}^2 + \tau^{(k)}\mathcal{R}(\mathbf{z}), \tag{6}$$

where $\lambda^{(k)}$ is the penalty term, and $\lambda^{(k)}$ and $\tau^{(k)}$ are learnable parameters independent for each stage. Then we merge the first two terms in Eq. (6) and reformulate it into following two subproblems:

$$\mathbf{u}^{(k)} = \mathbf{x}^{(k)} - \frac{1}{\lambda^{(k)}}\nabla f(\mathbf{x}^{(k)}), \tag{7}$$

$$\mathbf{x}^{(k+1)} \in \arg\min_{\mathbf{z}} \frac{1}{2}\|\mathbf{z} - \mathbf{u}^{(k)}\|_{\ell_2}^2 + \frac{\tau^{(k)}}{\lambda^{(k)}}\mathcal{R}(\mathbf{z}). \tag{8}$$

Specifically, Eq. (7) represents a gradient descent term:

$$\mathbf{u}^{(k)} = \mathbf{x}^{(k)} - \frac{1}{\lambda^{(k)}}\mathbf{A}^\top(\mathbf{A}\mathbf{x}^{(k)} - \mathbf{y}), \tag{9}$$

while Eq. (8) can be viewed as a denoising problem solvable using the proximal mapping operator. Here, we employ $\ell_1$-norm as the prior term to induce sparsity in the transformation domain, i.e., $\mathcal{R}(\mathbf{z}) = \|\mathbf{\Psi}\mathbf{z}\|_{\ell_1}$, where $\mathbf{\Psi} \in \mathbb{R}^{N \times N}$ denotes an orthonormal sparse basis. Thus, Eq. (8) can be reformulated as:

$$\mathbf{x}^{(k+1)} \in \arg\min_{\mathbf{z}} \frac{1}{2}\|\mathbf{z} - \mathbf{u}^{(k)}\|_{\ell_2}^2 + \frac{\tau^{(k)}}{\lambda^{(k)}}\|\mathbf{\Psi}\mathbf{z}\|_{\ell_1}. \tag{10}$$

**Theorem 1.** *Let $\mathbf{z} \in \mathbb{R}^N$, and let $\mathbf{\Psi} \in \mathbb{R}^{N \times N}$ be an orthonormal matrix, i.e., $\mathbf{\Psi}^T\mathbf{\Psi} = \mathbf{I}$, where $\mathbf{I}$ denotes the identity matrix. Then, Parseval's Theorem states that the Euclidean norm of $\mathbf{z}$ is equivalent to the Euclidean norm of its transform $\mathbf{\Psi}\mathbf{z}$, which can be mathematically expressed as:*

$$\|\mathbf{z}\|_{\ell_2}^2 = \mathbf{z}^T\mathbf{z} = (\mathbf{\Psi}\mathbf{z})^T(\mathbf{\Psi}\mathbf{z}) = \|\mathbf{\Psi}\mathbf{z}\|_{\ell_2}^2. \tag{11}$$

According to Theorem 1, the following can be derived:

$$\frac{1}{2}\|\mathbf{z} - \mathbf{u}^{(k)}\|_{\ell_2}^2 = \frac{1}{2}\|\mathbf{\Psi}(\mathbf{z} - \mathbf{u}^{(k)})\|_{\ell_2}^2 = \frac{1}{2}\|\mathbf{\Psi}\mathbf{z} - \mathbf{\Psi}\mathbf{u}^{(k)}\|_{\ell_2}^2. \tag{12}$$

By substituting Eq. (12) into Eq. (10), we arrive at the following expression:

$$\mathbf{x}^{(k+1)} \in \arg\min_{\mathbf{z}} \frac{1}{2}\|\mathbf{\Psi}\mathbf{z} - \mathbf{\Psi}\mathbf{u}^{(k)}\|_{\ell_2}^2 + \frac{\tau^{(k)}}{\lambda^{(k)}}\|\mathbf{\Psi}\mathbf{z}\|_{\ell_1}. \tag{13}$$

By differentiating Eq. (13) and setting the derivative equal to zero, we obtain:

$$(\mathbf{\Psi}\mathbf{z} - \mathbf{\Psi}\mathbf{u}^{(k)}) + \frac{\tau^{(k)}}{\lambda^{(k)}}\mathrm{sgn}(\mathbf{\Psi}\mathbf{z}) = 0. \tag{14}$$

Thus, it follows that:

$$\mathbf{\Psi}\mathbf{z} = \mathrm{soft}(\mathbf{\Psi}\mathbf{u}^{(k)}, \frac{\tau^{(k)}}{\lambda^{(k)}}), \tag{15}$$

Table 1: Average PSNR (dB) (upper) and SSIM (lower) performance comparisons of DUMoE and other ICS methods on various datasets at different sampling ratios (0.01, 0.04, 0.10 and 0.25).

| Methods | Urban100 | | | | | General100 | | | | | Set14 | | | | | McM18 | | | | |
|---|---|---|---|---|---|---|---|---|---|---|---|---|---|---|---|---|---|---|---|---|
| | 0.01 | 0.04 | 0.10 | 0.25 | Avg. | 0.01 | 0.04 | 0.10 | 0.25 | Avg. | 0.01 | 0.04 | 0.10 | 0.25 | Avg. | 0.01 | 0.04 | 0.10 | 0.25 | Avg. |
| ISTA-Net+ (CVPR 2018) | 16.67 0.3734 | 19.66 0.5370 | 23.51 0.7201 | 28.91 0.8834 | 22.19 0.6285 | 17.45 0.4131 | 21.56 0.624 | 26.49 0.8036 | 32.44 0.9237 | 24.49 0.6911 | 18.22 0.4014 | 22.08 0.5708 | 26.00 0.7289 | 30.62 0.8700 | 24.23 0.6428 | 19.99 0.4942 | 24.27 0.6577 | 28.54 0.8104 | 33.99 0.9237 | 26.70 0.7215 |
| AMP-Net (TIP 2021) | 19.62 0.5025 | 22.81 0.6825 | 26.04 0.8151 | 30.89 0.9202 | 24.84 0.7301 | 22.71 0.6282 | 26.96 0.7695 | 30.82 0.8722 | 36.01 0.9508 | 29.13 0.8052 | 21.64 0.5433 | 25.50 0.7007 | 28.77 0.8183 | 33.21 0.9144 | 27.28 0.7442 | 23.78 0.6426 | 27.90 0.7879 | 31.68 0.8860 | 36.88 0.9560 | 30.06 0.8181 |
| CASNet (TIP 2022) | 20.08 0.5366 | 23.73 0.7412 | 27.40 0.8606 | 32.19 0.9396 | 25.85 0.7695 | 23.48 0.6480 | 28.50 0.8171 | 32.78 0.9099 | 38.07 0.9657 | 30.71 0.8352 | 22.03 0.5600 | 26.04 0.7330 | 29.37 0.8467 | 33.95 0.9308 | 27.85 0.7676 | 24.23 0.6538 | 28.48 0.8166 | 32.47 0.9100 | 37.77 0.9659 | 30.74 0.8366 |
| DGUNet+ (CVPR 2022) | 20.15 0.5335 | 24.05 0.7478 | 28.01 0.8709 | 32.77 0.9452 | 26.25 0.7744 | 22.86 0.6190 | 27.92 0.8078 | 32.41 0.9073 | 37.55 0.9645 | 30.18 0.8247 | 21.86 0.5409 | 25.88 0.7250 | 29.34 0.8455 | 33.70 0.9294 | 27.69 0.7602 | 23.05 0.6267 | 28.16 0.8091 | 32.32 0.9070 | 37.74 0.9655 | 30.32 0.8271 |
| FSOINet (ICASSP 2022) | 19.87 0.5223 | 23.69 0.7376 | 27.53 0.8627 | 32.62 0.9430 | 25.93 0.7664 | 23.27 0.6363 | 28.39 0.8135 | 32.70 0.9085 | 38.13 0.9660 | 30.62 0.8311 | 22.00 0.5538 | 26.08 0.7324 | 29.35 0.8451 | 34.05 0.9309 | 27.87 0.7656 | 24.10 0.6464 | 28.50 0.8157 | 32.47 0.9097 | 37.85 0.9663 | 30.73 0.8345 |
| TransCS (TIP 2022) | 18.98 0.4398 | 23.27 0.7117 | 26.77 0.8418 | 31.77 0.9332 | 25.20 0.7316 | 21.66 0.5415 | 27.25 0.7843 | 31.39 0.8918 | 37.08 0.9604 | 29.34 0.7945 | 20.91 0.4853 | 25.50 0.7133 | 28.81 0.8343 | 33.37 0.9244 | 27.15 0.7393 | 22.81 0.5736 | 27.94 0.7976 | 31.88 0.8998 | 37.27 0.9627 | 29.98 0.8084 |
| AutoBCS (TCYB 2023) | 19.23 0.4991 | 22.50 0.7029 | 25.36 0.8242 | 29.60 0.9187 | 24.17 0.7362 | 22.24 0.6164 | 27.10 0.7964 | 30.76 0.8927 | 35.92 0.9581 | 29.00 0.8159 | 20.93 0.5343 | 25.07 0.7153 | 28.00 0.8286 | 32.14 0.9203 | 26.53 0.7496 | 23.26 0.6248 | 27.54 0.8002 | 31.13 0.8973 | 36.25 0.9608 | 29.55 0.8208 |
| SODAS-Net (TIM 2023) | 17.13 0.3947 | 20.85 0.5874 | 26.23 0.8084 | 31.86 0.9257 | 24.02 0.6791 | 19.52 0.5093 | 24.99 0.6998 | 30.58 0.8602 | 36.06 0.9454 | 27.79 0.7537 | 18.79 0.4349 | 23.19 0.6139 | 27.54 0.7812 | 32.39 0.8977 | 25.48 0.6819 | 20.84 0.5340 | 25.41 0.7079 | 30.16 0.8583 | 35.55 0.9434 | 27.99 0.7609 |
| TCS-Net (TCI 2023) | 19.61 0.4945 | 22.93 0.7036 | 25.87 0.8291 | 30.13 0.9241 | 24.64 0.7378 | 22.58 0.5978 | 26.57 0.7712 | 29.90 0.8748 | 34.63 0.9504 | 28.42 0.7986 | 21.64 0.5219 | 25.25 0.7073 | 28.19 0.8283 | 32.23 0.9206 | 26.83 0.7445 | 23.63 0.6144 | 27.54 0.7907 | 30.97 0.8913 | 35.89 0.9579 | 29.51 0.8136 |
| CSformer (TIP 2023) | 20.14 0.5298 | 24.03 0.7377 | 27.30 0.8483 | 31.83 0.9347 | 25.83 0.7626 | 23.35 0.6394 | 27.81 0.7986 | 31.60 0.8880 | 36.51 0.9558 | 29.82 0.8205 | 22.07 0.5493 | 25.87 0.7160 | 28.79 0.8214 | 32.95 0.9174 | 27.42 0.7510 | 23.66 0.6526 | 28.12 0.8030 | 31.69 0.8907 | 36.60 0.9570 | 30.02 0.8258 |
| OCTUF (CVPR 2023) | 19.88 0.5167 | 23.68 0.7328 | 27.79 0.8621 | 32.99 0.9445 | 26.08 0.7640 | 23.31 0.6346 | 28.35 0.8122 | 32.77 0.9084 | 38.26 0.9666 | 30.67 0.8305 | 21.94 0.5500 | 26.04 0.7302 | 29.47 0.8454 | 34.18 0.9312 | 27.91 0.7642 | 23.87 0.6409 | 28.33 0.8120 | 32.49 0.9093 | 37.93 0.9667 | 30.66 0.8322 |
| DPC-DUN (TIP 2023) | 17.31 0.4216 | 22.36 0.6768 | 26.96 0.8361 | 32.36 0.9323 | 24.75 0.7167 | 19.95 0.5363 | 26.61 0.7531 | 31.17 0.8716 | 36.50 0.9481 | 28.56 0.7773 | 19.04 0.4551 | 24.32 0.6630 | 28.03 0.7950 | 32.78 0.9023 | 26.04 0.7038 | 21.10 0.5553 | 26.51 0.7539 | 30.67 0.8701 | 35.86 0.9462 | 28.54 0.7814 |
| MTC-CSNet (TCYB 2023) | 19.63 0.4906 | 22.66 0.6858 | 25.81 0.8284 | 30.15 0.9228 | 24.56 0.7319 | 22.96 0.6122 | 27.26 0.7843 | 31.33 0.8970 | 36.33 0.9596 | 29.47 0.8133 | 21.68 0.5295 | 25.19 0.7018 | 28.47 0.8333 | 32.64 0.9226 | 27.00 0.7468 | 23.71 0.6227 | 27.62 0.7884 | 31.50 0.8999 | 36.68 0.9623 | 29.88 0.8183 |
| NesTD-Net (TIP 2024) | 20.13 0.5288 | 23.94 0.7432 | 27.80 0.8681 | 33.02 0.9448 | 26.22 0.7712 | 23.14 0.6165 | 28.58 0.8211 | 32.85 0.9123 | 38.42 0.9670 | 30.74 0.8292 | 22.32 0.5600 | 26.31 0.7393 | 29.62 0.8504 | 34.33 0.9330 | 28.15 0.7707 | 24.41 0.6535 | 28.70 0.8218 | 32.73 0.9125 | 37.98 0.9664 | 30.96 0.8385 |
| LTwIST (TCSVT 2024) | 19.46 0.4886 | 23.01 0.7061 | 26.76 0.8463 | 31.79 0.9349 | 25.26 0.7440 | 22.69 0.5989 | 27.53 0.7935 | 31.91 0.8990 | 37.31 0.9616 | 29.86 0.8133 | 21.49 0.5190 | 25.47 0.7112 | 28.88 0.8352 | 33.42 0.9249 | 27.31 0.7476 | 23.44 0.6108 | 27.64 0.7918 | 31.73 0.8995 | 36.97 0.9611 | 29.95 0.8158 |
| UFC-Net (CVPR 2024) | 19.69 0.5041 | 23.37 0.7195 | 27.55 0.8583 | 32.82 0.9423 | 25.86 0.7561 | 23.08 0.6145 | 27.92 0.7988 | 32.31 0.9014 | 37.75 0.9624 | 30.27 0.8193 | 21.79 0.5324 | 25.67 0.7163 | 29.10 0.8363 | 33.81 0.9259 | 27.59 0.7527 | 23.73 0.6240 | 27.95 0.7984 | 31.97 0.9011 | 37.24 0.9619 | 30.22 0.8214 |
| **DUMoE (Our Method)** | **20.33 0.5420** | **24.48 0.7614** | **28.43 0.8773** | **33.42 0.9481** | **26.67 0.7822** | **24.02 0.6545** | **28.96 0.8261** | **33.15 0.9149** | **38.45 0.9676** | **31.15 0.8408** | **22.40 0.5643** | **26.44 0.7407** | **29.83 0.8516** | **34.42 0.9334** | **28.27 0.7725** | **24.42 0.6562** | **28.85 0.8245** | **32.90 0.9146** | **38.09 0.9676** | **31.07 0.8407** |

where soft denotes the soft thresholding function, defined as $\mathrm{soft}(\mathbf{x}, \theta) \equiv \mathrm{sgn}(\mathbf{x}) \max\{|\mathbf{x}| - \theta, 0\}$. Consequently, the closed-form solution of Eq. (13) is given by:

$$\mathbf{x}^{(k+1)} = \mathbf{\Psi}^T \mathrm{soft}\left(\mathbf{\Psi}\mathbf{u}^{(k)}, \frac{\tau^{(k)}}{\lambda^{(k)}}\right). \tag{16}$$

However, obtaining $\mathbf{x}^{(k+1)}$ in Eq. (10) remains challenging when $\mathbf{\Psi}$ is non-orthogonal or represents a nonlinear transform (Zhang & Ghanem (2018)). To address this, we substitute $\mathbf{\Psi}$ with a learnable, deep learning-based structure $\mathcal{D}$, as presented in Eq. (17), which allows for learning a sparse representation of $\mathbf{z}$, enhancing both model flexibility and adaptability.

$$\mathbf{x}^{(k+1)} = \widetilde{\mathcal{D}}\left(\mathrm{soft}\left(\mathcal{D}(\mathbf{u}^{(k)}), \frac{\tau^{(k)}}{\lambda^{(k)}}\right)\right), \tag{17}$$

where $\widetilde{\mathcal{D}}$ denotes the left inverse of $\mathcal{D}$. Here, both $\mathcal{D}$ and $\widetilde{\mathcal{D}}$ are depth-wise convolutions with a $3 \times 3$ kernel. It is worth noting that the image-level feature transmission in DUNs often results in information loss (Zhang et al. (2022); Song et al. (2023c)) during the reconstruction. Therefore, we use the Sigmoid function and residual connections to achieve the weighted feature fusion and obtain the output of the DUSE, denoted as $\widetilde{\mathbf{x}}^{(k+1)}$, in the $k$-th iteration stage.

## 3.3 LOSS FUNCTION

We adopt different loss functions, denoted as $\mathcal{L}_{\text{deviation}}$, to quantify the deviation between the reconstructed image and the corresponding ground truth image for various CI tasks. For instance, we utilize the $\ell_2$-norm loss for ICS and CS-MRI tasks, and the Charbonnier loss (Charbonnier et al. (1994)) for SCI tasks. Furthermore, to promote load balance and competition across different DUSE (Fedus et al. (2022)), we employ the coefficient of variation to measure the dispersion of gate scores of DUSE in each iteration stage:

$$\mathcal{L}_{C_v} = \frac{1}{n} \sum_{k=1}^{n} \left(\frac{\mathrm{std}(w^{(k)})}{\mathrm{mean}(w^{(k)})}\right)^2, \tag{18}$$

where $n$ denotes the number of iteration stages and $w^{(k)} = \{w_1^{(k)}, w_2^{(k)}, w_3^{(k)}\}$ represents the gate scores in the $k$-th iteration stage. Consequently, the loss function of DUMoE is formulated as follows:

$$\mathcal{L}_{\text{total}} = \mathcal{L}_{\text{deviation}} + \eta \mathcal{L}_{C_v}, \tag{19}$$

where $\eta$ is the weight of $\mathcal{L}_{C_v}$. In our experiments, we set $\eta$ to $1 \times 10^{-3}$.

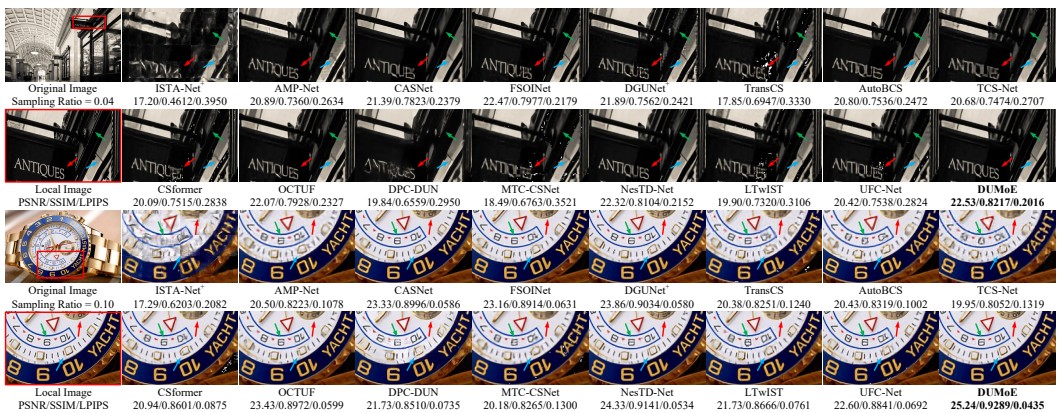

Figure 3: Comparisons of visual results and corresponding PSNR (dB)/SSIM/LPIPS (Zhang et al. (2018)) performance between DUMoE and other advanced ICS methods at sampling ratios of 0.04 and 0.10. Key details are highlighted with arrows. Please zoom in for better comparisons.

## 4 EXPERIMENTS

In this section, we conduct extensive experiments across three CI tasks: ICS, CS-MRI, and SCI. We set the default number of iteration stages to $n = 5$ and corresponding channels to $C_1 = 32$, $C_2 = 48$, and $C_3 = 80$. Besides, we highlight the best and second-best results in the tables using red and blue colors, respectively. Further implementation details for each CI task are provided in Appendix A.4. Additionally, more experiments as well as supplementary visualizations are available in Appendix A.5.

### 4.1 NATURAL IMAGE COMPRESSIVE SENSING

We conduct qualitative comparisons between DUMoE and sixteen ICS methods, including ISTA-Net[+] (Zhang & Ghanem (2018)), AMP-Net (Zhang et al. (2021)), CASNet (Chen & Zhang (2022)), DGUNet[+] (Mou et al. (2022)), FSOINet (Chen et al. (2022)), TransCS (Shen et al. (2022)), AutoBCS (Gan et al. (2023a)), SODAS-Net (Song & Zhang (2023)), TCS-Net (Gan et al. (2023b)), CSformer (Ye et al. (2023)), OCTUF (Song et al. (2023c)), DPC-DUN (Song et al. (2023b)), MTC-CSNet (Shen et al. (2024)), NesTD-Net (Gan et al. (2024a)), LTwIST (Gan et al. (2024b)), and UFC-Net (Wang & Gan (2024)), across four widely-used benchmark datasets: Urban100 (Huang et al. (2015)), General100 (Dong et al. (2016)), Set14 (Zeyde et al. (2012)), and McM18 (Zhang et al. (2011)).

Tab. 1 demonstrates that DUMoE consistently outperforms other methods in terms of PSNR and SSIM across all tested datasets and various sampling ratios. Specifically, on the Urban100 at a sampling ratio of 0.10, DUMoE surpasses OCTUF, LTwIST, DPC-DUN, MTC-CSNet, NesTD-Net and UFC-Net by approximately 0.64 dB (2.30%), 1.67 dB (6.24%), 1.47 dB (5.45%), 2.62 dB (10.15%), 0.63 dB (2.27%), and 0.88 dB (3.19%) in terms of PSNR, respectively. Similarly, regarding SSIM, DUMoE leads by around 0.0152 (1.76%), 0.0310 (3.66%), 0.0412 (4.93%), 0.0489 (5.90%), 0.0092 (1.06%), and 0.0190 (2.21%), respectively. Moreover, Fig. 3 shows that DUMoE consistently achieves superior performance in terms of human perception quality compared to methods such as NesTD-Net, LTwIST, UFC-Net, and others. Even at low sampling ratios of 0.04 and 0.10, DUMoE excels in recovering fine-grained image details with reduced noise, distortion, blurring, and absence of blocking artifacts. This underscores the effectiveness of DUMoE in reconstructing images with higher human perception quality and overall image quality. For additional experiments at high sampling ratios, please refer to Appendix A.5.1.

### 4.2 COMPRESSIVE SENSING MRI

As shown in Tab. 2, we compare DUMoE with eleven CS-MRI methods, including ISTA-Net[+] (Zhang & Ghanem (2018)), RDN (Sun et al. (2018)), DC-CNN (Schlemper et al. (2018)), CDDN (Zheng et al. (2019)), ADMM-CSNet (Yang et al. (2020)), NESTANets (Neyra-Nesterenko & Adcock (2022)),

Table 2: Average PSNR (dB) and SSIM performance comparisons of DUMoE and other CS-MRI methods on Brain dataset at various sampling ratios (0.05, 0.10, 0.20, 0.30 and 0.40).

| Methods | 0.05 | | 0.10 | | 0.20 | | 0.30 | | 0.40 | | Avg. | |
|---|---|---|---|---|---|---|---|---|---|---|---|---|
| | PSNR | SSIM | PSNR | SSIM | PSNR | SSIM | PSNR | SSIM | PSNR | SSIM | PSNR | SSIM |
| Zero-filled | 24.20 | 0.5417 | 26.81 | 0.6030 | 30.41 | 0.7229 | 33.01 | 0.8023 | 35.14 | 0.8568 | 29.91 | 0.7053 |
| ISTA-Net$^+$ (CVPR 2018) | 31.28 | 0.8547 | 34.62 | 0.9035 | 38.57 | 0.9478 | 40.90 | 0.9631 | 42.62 | 0.9724 | 37.60 | 0.9283 |
| RDN (AAAI 2018) | 30.95 | 0.8421 | 34.38 | 0.8998 | 38.47 | 0.9474 | 40.82 | 0.9630 | 42.50 | 0.9719 | 37.42 | 0.9248 |
| DC-CNN (TMI 2018) | 30.81 | 0.8370 | 34.33 | 0.8957 | 38.43 | 0.9467 | 40.53 | 0.9526 | 42.02 | 0.9717 | 37.22 | 0.9207 |
| CDDN (NeurIPS 2019) | 31.58 | 0.8513 | 34.67 | 0.9014 | 38.65 | 0.9476 | 40.95 | 0.9633 | 42.74 | 0.9731 | 37.72 | 0.9273 |
| ADMM-CSNet (TPAMI 2020) | 31.37 | 0.8608 | 34.45 | 0.8985 | 38.52 | 0.9471 | 40.81 | 0.9629 | 42.71 | 0.9729 | 37.57 | 0.9284 |
| NESTANets (STSPDA 2022) | 26.65 | 0.6044 | 30.79 | 0.7670 | 35.20 | 0.8866 | 38.07 | 0.9314 | 40.16 | 0.9523 | 34.17 | 0.8283 |
| HiTDUN (J-STSP 2022) | 32.72 | 0.8770 | 35.35 | 0.9104 | 39.02 | 0.9510 | 41.21 | 0.9651 | 42.87 | 0.9737 | 38.23 | 0.9354 |
| PUERT (J-STSP 2022) | 31.51 | 0.8542 | 34.84 | 0.9068 | 38.78 | 0.9495 | 41.01 | 0.9642 | 42.73 | 0.9732 | 37.77 | 0.9296 |
| LTwIST (TCSVT 2024) | 31.30 | 0.8536 | 34.11 | 0.9043 | 36.68 | 0.9361 | 39.46 | 0.9523 | 41.47 | 0.9663 | 36.60 | 0.9225 |
| NesTD-Net (TIP 2024) | 33.71 | 0.8934 | 36.15 | 0.9243 | 39.43 | 0.9536 | 41.32 | 0.9658 | 42.90 | 0.9740 | 38.70 | 0.9422 |
| UFC-Net (CVPR 2024) | 32.63 | 0.8779 | 34.68 | 0.9064 | 38.85 | 0.9502 | 41.04 | 0.9644 | 42.73 | 0.9732 | 37.99 | 0.9344 |
| **DUMoE (Our Method)** | **34.28** | **0.9047** | **36.39** | **0.9274** | **39.65** | **0.9555** | **41.57** | **0.9668** | **43.11** | **0.9746** | **39.00** | **0.9458** |

Table 3: PSNR (dB)/SSIM performance comparisons of DUMoE and other SCI methods on 10 simulation scenes.

| Methods | Scene1 | Scene2 | Scene3 | Scene4 | Scene5 | Scene6 | Scene7 | Scene8 | Scene9 | Scene10 | Avg. |
|---|---|---|---|---|---|---|---|---|---|---|---|
| GAP-TV (ICIP 2017) | 28.16/0.913 | 23.90/0.818 | 20.44/0.762 | 23.33/0.872 | 28.11/0.920 | 27.69/0.885 | 20.62/0.811 | 24.68/0.831 | 22.84/0.800 | 23.02/0.843 | 24.28/0.845 |
| DeSCI (TPAMI 2019) | 28.30/0.910 | 27.46/0.901 | 31.98/0.955 | 32.56/0.971 | 28.06/0.933 | 27.43/0.914 | 25.51/0.945 | 24.51/0.876 | 31.80/0.935 | 22.29/0.822 | 27.99/0.916 |
| Lambda-Net (ICCV 2019) | 29.71/0.830 | 27.70/0.742 | 29.53/0.846 | 37.53/0.911 | 26.60/0.790 | 27.25/0.787 | 26.61/0.782 | 26.20/0.781 | 28.54/0.798 | 26.14/0.701 | 28.58/0.797 |
| ADMM-Net (ICCV 2019) | 34.09/0.924 | 33.58/0.904 | 35.02/0.935 | 41.24/0.972 | 31.79/0.926 | 32.52/0.929 | 32.38/0.901 | 30.68/0.912 | 33.70/0.921 | 30.64/0.905 | 33.56/0.923 |
| TSA-Net (ECCV 2020) | 32.31/0.898 | 31.07/0.863 | 32.30/0.918 | 39.53/0.959 | 29.44/0.887 | 31.06/0.905 | 30.26/0.883 | 29.31/0.893 | 31.62/0.912 | 29.20/0.867 | 31.61/0.899 |
| DGSMP (CVPR 2021) | 33.35/0.920 | 31.66/0.892 | 32.93/0.925 | 40.39/0.970 | 29.46/0.894 | 32.74/0.938 | 31.14/0.898 | 31.32/0.932 | 31.53/0.925 | 31.51/0.934 | 32.60/0.923 |
| MST-L (CVPR 2022) | 35.43/0.946 | 36.11/0.949 | 36.39/0.955 | 42.05/0.977 | 32.94/0.950 | 34.71/0.957 | 34.08/0.932 | 32.88/0.953 | 35.04/0.947 | 32.74/0.946 | 35.24/0.951 |
| HDNet (CVPR 2022) | 35.10/0.940 | 35.65/0.943 | 36.04/0.948 | 42.47/0.978 | 32.67/0.950 | 34.46/0.956 | 33.64/0.930 | 32.43/0.948 | 34.86/0.947 | 32.34/0.943 | 34.97/0.948 |
| CST-L+ (ECCV 2022) | 35.87/0.954 | 36.84/0.958 | 38.20/0.966 | 42.53/0.982 | 33.11/0.958 | 35.76/0.967 | 34.73/0.947 | 34.33/0.967 | 36.31/0.961 | 33.04/0.952 | 36.07/0.961 |
| BiSRNet (NeurIPS 2023) | 30.87/0.853 | 29.22/0.795 | 28.97/0.830 | 35.87/0.909 | 28.20/0.828 | 30.19/0.860 | 27.81/0.806 | 28.71/0.845 | 29.39/0.834 | 27.84/0.810 | 29.71/0.837 |
| RDFNet (TCI 2023) | 33.40/0.950 | 32.38/0.904 | 34.47/0.961 | 37.70/0.976 | 32.67/0.957 | 35.80/0.963 | 27.67/0.939 | 33.09/0.956 | 34.66/0.958 | 31.54/0.949 | 33.34/0.956 |
| GAP-Net (IJCV 2023) | 33.62/0.926 | 30.08/0.914 | 33.07/0.944 | 40.94/0.966 | 30.77/0.925 | 33.60/0.936 | 27.41/0.915 | 31.25/0.918 | 33.56/0.937 | 30.36/0.914 | 32.47/0.929 |
| EDUNet (NN 2024) | 36.48/0.951 | 37.65/0.961 | 37.19/0.963 | 42.85/0.981 | 34.29/0.962 | 35.70/0.966 | 35.37/0.949 | 34.18/0.962 | 36.81/0.960 | 33.46/0.951 | 36.40/0.961 |
| DWMT (AAAI 2024) | 36.46/0.957 | 37.75/0.963 | 38.47/0.965 | 44.23/0.984 | 33.99/0.963 | 36.17/0.970 | 35.22/0.949 | 34.56/0.968 | 37.41/0.965 | 34.00/0.959 | 36.83/0.964 |
| **DUMoE (Our Method)** | **36.73/0.959** | **38.87/0.971** | **40.46/0.974** | **45.69/0.989** | **34.87/0.969** | **36.58/0.973** | **35.88/0.952** | **34.78/0.971** | **38.79/0.971** | **33.74/0.959** | **37.64/0.969** |

HiTDUN (Zhang et al. (2022)), PUERT (Xie et al. (2022)), NesTD-Net (Gan et al. (2024a)), LTwIST (Gan et al. (2024b)), and UFC-Net (Wang & Gan (2024)) on the widely-used Brain dataset (Yang et al. (2020)) using Pseudo Radial masks as sub-sampling matrix. Specifically, at a sampling ratio of 0.05, DUMoE significantly outperforms NesTD-Net, LTwIST, and UFC-Net, with improvements of approximately 0.57 dB (1.69%), 2.98 dB (9.52%), and 1.65 dB (5.06%) in PSNR, respectively, and leads by around 0.0113 (1.26%), 0.0511 (5.99%), and 0.0268 (3.05%) in terms of SSIM, respectively. For additional visualizations, please refer to Appendix A.5.2.

## 4.3 SNAPSHOT COMPRESSIVE IMAGING

We perform qualitative comparisons between DUMoE and fourteen SCI methods, namely GAP-TV (Yuan (2016)), DeSCI (Liu et al. (2019)) Lambda-Net (Miao et al. (2019)), ADMM-Net (Ma et al. (2019)), TSA-Net (Meng et al. (2020)), DGSMP (Huang et al. (2021)), MST-L (Cai et al. (2022b)), HDNet (Hu et al. (2022)), CST-L$^+$ (Cai et al. (2022a)), BiSRNet (Cai et al. (2023)), RDFNet (Zhou et al. (2023)), GAP-Net (Meng et al. (2023)), EDUNet (Qin et al. (2024)) and DWMT (Luo et al. (2024)), using widely-used ten scenes from KAIST dataset (Choi et al. (2017)). As shown in Tab. 3, when compared to GAP-Net, RDFNet, EDUNet, and DWMT, DUMoE achieves an average PSNR improvement of approximately 5.17 dB (15.92%), 4.30 dB (12.90%), 1.24 dB (3.41%), and 0.81 dB (2.20%) across the ten scenes, respectively. Moreover, in terms of average SSIM on ten scenes, DUMoE maintains a lead of approximately 0.040 (4.31%), 0.013 (1.36%), 0.008 (0.83%), and 0.005 (0.52%), respectively. For additional visualizations and experiments on real HSI data, please refer to Appendix A.5.3.

## 5 DISCUSSION

In this section, we delve into several discussions concerning DUMoE, primarily based on ICS experiments, but the insights are equally applicable to other tasks as well.

Table 4: Ablation studies on different cases (a) and number of iteration stages (b), as well as complexity analysis of various methods (c).

(a) PSNR (dB), SSIM, parameters (M) and FLOPs (G) for different ablation cases on various datasets at a sampling ratio of 0.25.

| Cases | Urban100 | | General100 | | Set14 | | McM18 | | Params. | FLOPs |
|---|---|---|---|---|---|---|---|---|---|---|
| | PSNR | SSIM | PSNR | SSIM | PSNR | SSIM | PSNR | SSIM | | |
| w/o SR | 33.39 | 0.9479 | 38.41 | 0.9675 | 34.39 | 0.9329 | 38.12 | 0.9675 | 4.17 | 158.72 |
| w/o DUSE | 33.30 | 0.9469 | 38.34 | 0.9674 | 34.35 | 0.9333 | 38.03 | 0.9675 | 4.43 | 150.53 |
| w/o MSGate | 31.15 | 0.9289 | 37.34 | 0.9629 | 33.24 | 0.9252 | 37.39 | 0.9642 | 1.18 | 92.77 |
| w/o DAM | 33.21 | 0.9465 | 38.30 | 0.9672 | 34.29 | 0.9325 | 38.02 | 0.9670 | 3.92 | 116.74 |
| DUMoE | 33.42 | 0.9481 | 38.45 | 0.9676 | 34.42 | 0.9334 | 38.09 | 0.9676 | 4.17 | 142.34 |

(b) PSNR (dB)/SSIM, parameters (M) and FLOPs (G) for different number of stages in DUMoE on General100 at a sampling ratio of 0.10.

| Stages | 3 | 4 | 5 (default) | 7 | 5 (w/o share weights) |
|---|---|---|---|---|---|
| PSNR/SSIM | 32.88/0.9128 | 33.00/0.9132 | 33.15/0.9149 | 33.24/0.9151 | 32.98/0.9139 |
| Params. | 4.01 | 4.01 | 4.01 | 4.01 | 5.87 |
| FLOPs | 91.18 | 117.76 | 141.31 | 190.46 | 141.31 |

(c) Comparisons of various resources required by different methods on a 256×256 image at a sampling ratio of 0.10.

| Methods | Params. (M) | FLOPs (G) | Inference time (ms) | Inference memory (MB) | Model size (MB) |
|---|---|---|---|---|---|
| CASNet | 16.90 | 205.24 | 33±2 | 1652 | 64.77 |
| DGUNet[+] | 6.81 | 97.79 | 27±1 | 1124 | 26.61 |
| FSOINet | 0.64 | 17.19 | 8±2 | 852 | 2.53 |
| TransCS | 1.49 | 25.86 | 187±26 | 515 | 20.43 |
| AutoBCS | 2.01 | 20.11 | 26±3 | 651 | 7.72 |
| SODAS-Net | 0.92 | 64.69 | 10±7 | 720 | 3.54 |
| TCS-Net | 0.52 | 7.04 | 5±3 | 1553 | 3.23 |
| OCTUF | 0.40 | 21.51 | 16±6 | 824 | 1.67 |
| DPC-DUN | 1.64 | 65.54 | 25±5 | 579 | 6.43 |
| MTC-CSNet | 0.92 | 20.61 | 10±0 | 605 | 3.61 |
| NesTD-Net | 5.57 | 347.92 | 82±12 | 1957 | 21.38 |
| LTwIST | 23.49 | 110.46 | 103±6 | 707 | 89.99 |
| UFC-Net | 1.74 | 115.58 | 84±28 | 844 | 7.19 |
| **DUMoE** | 4.01 | 141.31 | 78±6 | 1745 | 15.46 |

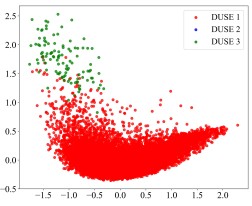

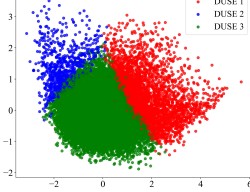

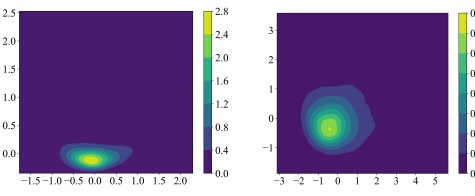

(a) PCA visualization of hidden states in w/o MSGate.

(b) PCA visualization of hidden states in DUMoE.

(c) Gaussian KDE visualization of hidden states in w/o MSGate.

(d) Gaussian KDE visualization of hidden states in DUMoE.

Figure 4: Analysis of the representation collapse of the hidden states in MSGate of DUMoE. (a) and (b) illustrate the spatial structure of the experts using Principal Component Analysis (PCA), where each data point represents an image to be routed, and its color corresponds to the assigned DUSE. (c) and (d) show the diversity of these hidden states, computed using Gaussian Kernel Density Estimation (KDE) and visualized as heatmaps.

## 5.1 ABLATION STUDY

**Different components**. We perform ablation experiments to evaluate the contributions of switch routing strategy, DUSE, MSGate, and DAM within the DUMoE architecture. Specifically, we obtain the ablation cases by adopting all the experts instead of switch routing (w/o SR), replacing DUSE with two $3 \times 3$ convolutions and GELU activation in between (w/o DUSE), replacing MSGate with a linear layer (w/o MSGate), and removing DAM from DA-SA (w/o DAM).

First, as shown in Tab. 4a, DUMoE achieves better or comparable performance with the same parameter counts and fewer Floating Point Operations (FLOPs) compared to w/o SR, while also achieving superior performance with fewer parameters and FLOPs compared to w/o DUSE, demonstrating the effectiveness of the switch routing strategy and our proposed DUSE.

Furthermore, as illustrated in Fig. 4, we visualize and analyze the hidden states (i.e., the input features of the gate module) in the MSGate of DUMoE from the aspect of representation collapse (Chi et al. (2022)). We employ images from the CIFAR-10 and CIFAR-100 test sets (Krizhevsky (2009)), comprising a total of 20,000 images, for visualizations and analyses.

Initially, we use Principal Component Analysis (PCA) to extract the first two principal components from the hidden states. As illustrated in Fig. 4a and Fig. 4b, each data point represents an image to be routed, with its color corresponding to the assigned DUSE. In Fig. 4a, the points are predominantly mixed together, indicative of unbalanced routing. Conversely, in Fig. 4b, DUMoE exhibits a well-structured feature space with clear cluster distinctions, suggesting successful projection of images by our MSGate while preserving routing features.

Subsequently, we apply Gaussian Kernel-Density Estimation (KDE) to the hidden states processed by PCA, using the Scott method as the bandwidth estimator. Compared to Fig. 4c, Fig. 4d showcases uniformly distributed hidden states, indicating balanced expert assignment and reduced representation collapse (Chi et al. (2022)).

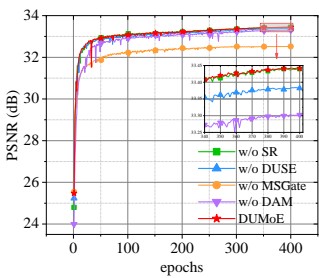 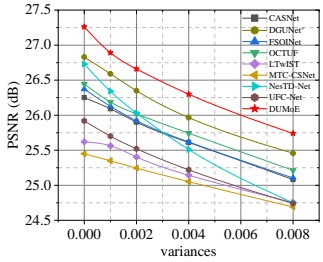 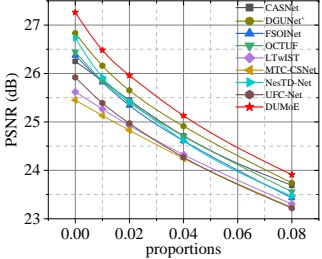

(a) PSNR (dB) of different ablation cases on the validation set at a sampling ratio of 0.25 during the training epochs.

(b) PSNR (dB) of different methods on Set11 at a sampling ratio of 0.04 under different Gaussian noise levels.

(c) PSNR (dB) of different methods on Set11 at a sampling ratio of 0.04 under different pepper-and-salt noise levels.

Figure 5: PSNR (dB) for different ablation cases on validation set during the training epochs (a), different levels of Gaussian noise (b) and pepper-and-salt noise (c) for various methods on Set11.

Finally, as shown in Fig. 5a, compared to other cases, w/o DAM converges more slowly in the initial epochs and does not perform as well as DUMoE in the final convergence. This demonstrates that DAM effectively guides DUMoE to focus on degraded image areas while enhancing attention to crucial details, thus improving feature extraction capabilities. Besides, we provide more visual analyses of the DAM and feature maps at each stage in Appendix A.5.4 and Appendix A.5.5, respectively.

**Number of iteration stages**. We explore the influence of varying the number of iteration stages in DUMoE, specifically examining configurations with 3, 4, 5 (default), and 7 stages. As presented in Tab. 4b, due to the weight sharing across intermediate stages, DUMoE's performance is observed to scale with FLOPs without increasing the parameter counts, underscoring the effectiveness of our iterative network design. Moreover, compared to the case of w/o sharing weights, DUMoE achieves superior performance with same FLOPs and fewer parameters.

## 5.2 COMPLEXITY ANALYSIS

We conduct a thorough analysis of computational efficiency and hardware utilization for DUMoE and other methods on a 256×256 image with a sampling ratio of 0.10 on the RTX 4090 GPU. Average inference time (ms) and its standard deviation are computed over 500 passes, with memory consumption measured using the nvidia-smi command. Model size reflects the storage requirements of each model, along with reported parameter counts and FLOPs for comparison. As indicated in Tab. 4c, when compared to CASNet, DGUNet$^+$, and NesTD-Net—each achieving several second-best results in Tab. 1—our proposed DUMoE demonstrates superior performance while featuring fewer parameters than CASNet, DGUNet$^+$, and NesTD-Net, as well as fewer FLOPs than CASNet and NesTD-Net. Notably, in comparison to NesTD-Net, DUMoE achieves superior performance while reducing parameters by 1.56 M (28.01%), and FLOPs by 206.61 G (59.38%).

## 5.3 PERFORMANCE UNDER NOISE

We assess the robustness of our DUMoE under various levels of Gaussian and pepper-and-salt noise to demonstrate its effectiveness. Specifically, we introduce four levels of Gaussian noise variances (0.001, 0.002, 0.004, and 0.008) and pepper-and-salt proportions (0.01, 0.02, 0.4, and 0.08) to the Set11 (Kulkarni et al. (2016)) and evaluate the model's performance on these noisy images. As shown in Fig. 5b and Fig. 5c, while the performance of each method declines with increasing noise levels, DUMoE consistently outperforms the other methods across all tested noise levels.

## 6 CONCLUSION

In this paper, we propose DUMoE, a novel sparse Deep Unfolding MoE framework for CI tasks. DUMoE addresses key challenges in CI recovery problems by integrating innovative components: the DAM, MSGate, and DUSE. Notably, our work represents the first attempt to study deep unfolding paradigm within the MoE framework. Extensive experiments across various CI tasks, including ICS, CS-MRI and SCI, demonstrate the superior performance and effectiveness of our proposed DUMoE.

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

# A    APPENDIX

In this appendix, we provide more details not covered in the main paper, including:

- Introduction of SpaRSA in Appendix A.1.
- Mathematical descriptions of sampling process for different CI tasks in Appendix A.2;
- Mathematical descriptions of initialization process for different CI tasks in Appendix A.3;
- Implementation specifics of various experiments in Appendix A.4.
- Additional experiments and visualizations in Appendix A.5;
- Limitations of our work in Appendix A.6;
- Code submission and reproducibility in Appendix A.7.

## A.1    SPARSA

SpaRSA (Wright et al. (2009)) (short for **Spa**rse **R**econstruction by **S**eparable **A**pproximation) is a general approach for solving unconstrained optimization problem as follows:

$$\min_{\mathbf{x}} \ \theta(\mathbf{x}) := f(\mathbf{x}) + \tau\gamma(\mathbf{x}), \tag{20}$$

where $f$ is a smooth function, $\tau$ is the regularization parameter and $\gamma$ is always non-smooth and non-convex, which is usually called regularization function and is finite for all $\mathbf{x} \in \mathbb{R}^N$. Specifically, SpaRSA solve Eq. (20) by iterating following equations:

$$\mathbf{x}^{(k+1)} \in \arg\min_{\mathbf{z}} (\mathbf{z} - \mathbf{x}^{(k)})\nabla f(\mathbf{x}^{(k)}) + \frac{\lambda}{2}\|\mathbf{z} - \mathbf{x}\|_{\ell 2}^2 + \tau\gamma(\mathbf{z}), \tag{21}$$

where $k$ denotes the $k$-th iterations and $\lambda \in \mathbb{R}^+$. Notably, the choice of $\gamma$ can vary, including $\ell_0$-norm, $\ell_1$-norm, total-variation norm, etc., for different applications, such as image processing and restoration problems. Compared with algorithms that are specially designed for particular tasks, such as ISTA

and FPC, SpaRSA serves as an effective and versatile approach to handle these problems and is computationally competitive. However, being a traditional iterative optimization algorithm, SpaRSA still requires hand-crafted parameter setting, such as $\lambda$ and $\tau$ for different tasks. By incorporating DL modules into SpaRSA, we can fully exploit the potential of powerful generalization ability in SpaRSA with the fast feature learning and correspondence capabilities of DL. Thus, we introduce the deep unfolding SpaRSA as the experts in our proposed DUMoE framework.

## A.2 SAMPLING PROCESS FOR DIFFERENT CI TASKS

**ICS**: Given an input natural image $\mathbf{x} \in \mathbb{R}^{H \times W}$ with height and width of $H$ and $W$, respectively, image $\mathbf{x}$ is initially partitioned into non-overlapping blocks of size $B \times B$. In cases where the width or height of $\mathbf{x}$ is not perfectly divisible by $B$, zero-padding is employed to ensure uniform block sizes. These blocks are then transformed into vectors, and a sampling matrix $\mathbf{A} \in \mathbb{R}^{M \times N}$ ($M \ll N$) is applied to yield measurement $\mathbf{y} \in \mathbb{R}^M$. In our ICS experiments, $\mathbf{A}$ is initialized as Gaussian matrix. Let $\mathcal{F}_{\text{vec}}(\cdot) : \mathbb{R}^{W \times H} \to \mathbb{R}^{B^2}$ denote the partitioning and vectorization function, and $\sigma$ represent the sampling ratio, where $M = \lfloor N \times \sigma \rfloor = \lfloor B^2 \times \sigma \rfloor$. The sampling process in ICS can be represented as follows:

$$\mathbf{y} = \mathbf{A}\mathcal{F}_{\text{vec}}(\mathbf{x}). \tag{22}$$

**CS-MRI**: CS-MRI employs a partial Fourier transform matrix as the sampling matrix $\mathbf{A} = \mathbf{U}\mathbf{F}$, where $\mathbf{U}$ represents a sub-sampling mask, and $\mathbf{F}$ corresponds to the Discrete Fourier Transform (DFT). In our CS-MRI experiments, we adopt the Pseudo Radial masks as $\mathbf{U}$. Additionally, the size of $\mathbf{U}$ matches that of the input image $\mathbf{x}$, and $\sigma$ denotes the ratio between the number of measurement points $M$ and the total number of pixels $N$ in $\mathbf{x}$, i.e., $\sigma = \frac{M}{N}$. The sampling process of CS-MRI is mathematically represented as:

$$\mathbf{y} = \mathbf{A}\mathbf{x} = \mathbf{U}\mathbf{F}\mathbf{x}. \tag{23}$$

**SCI**: Consider a 3D hyperspectral image (HSI) $\mathbf{x} \in \mathbb{R}^{H \times W \times N_\zeta}$, where $W$, $H$, and $N_\zeta$ represent its width, height, and number of wavelengths, respectively. The process begins with the application of a pre-defined coded aperture $\mathbf{A}_\zeta \in \mathbb{R}^{H \times W}$ to modulate the captured HSI, resulting in the transformed HSI denoted as $\mathbf{x}'$:

$$\mathbf{x}'(:, :, n_\zeta) = \mathbf{x}(:, :, n_\zeta) \odot \mathbf{A}_\zeta, \tag{24}$$

where $\odot$ denotes the Hadamard product, and $n_\zeta \in [1, ..., N_\zeta]$ denotes the spectral channels. After modulation, the modulated HSI $\mathbf{x}'$ is subjected to spatial shifts through a disperser, resulting in a transformed measurement $\mathbf{x}'' \in \mathbb{R}^{H \times (W + d(N_\zeta - 1))}$. This process induces shear and tilt effects, where $d$ denotes the step of spatial shifting. The dispersion operation can be expressed as:

$$\mathbf{x}''(u, v, n_\zeta) = \mathbf{x}'(x, y + d(\zeta_n - \zeta_c), n_\zeta). \tag{25}$$

Here, $\zeta_c$ represents the reference wavelength, $\zeta_n$ signifies the wavelength of the $n_\zeta$-th spectral channel, $(u, v)$ denotes the coordinate system on the detector array, and $d(\zeta_n - \zeta_c)$ characterizes the spatial shifting offset of the $n_\zeta$-th channel on $\mathbf{x}''$. As a result, the 2D compressed measurement $\mathbf{y} \in \mathbb{R}^{H \times (W + d(N_\zeta - 1))}$ is acquired through the following summation operation:

$$\mathbf{y} = \sum_{n_\zeta = 1}^{N_\zeta} \mathbf{x}''(:, :, n_\zeta) + \mathbf{E}. \tag{26}$$

Here, $\mathbf{E}$ signifies the random image noise produced by the photon sensing detector.

## A.3 INITIALIZATION PROCESS FOR DIFFERENT CI TASKS

**ICS**: For initialization, a matrix multiplication is performed using the transpose of the sampling matrix $\mathbf{A}^T \in \mathbb{R}^{N \times M}$ and $\mathbf{y}$ to obtain a vector of image blocks. Following this, the function $\widetilde{\mathcal{F}}_{\text{vec}}(\cdot) : \mathbb{R}^{B^2} \to \mathbb{R}^{W \times H}$ is applied to recover the image blocks and assemble them into the initial estimate $\mathbf{x}^{(0)}$. This initialization process can be represented as:

$$\mathbf{x}^{(0)} = \widetilde{\mathcal{F}}_{\text{vec}}(\mathbf{A}^T \mathbf{y}). \tag{27}$$

**CS-MRI**: As for initialization, we apply the inverse DFT, denoted as $\widetilde{\mathbf{F}}$, to the acquired measurement $\mathbf{y}$ to obtain the initial estimate $\mathbf{x}^{(0)}$. Therefore, the initialization of CS-MRI can be expressed as:

$$\mathbf{x}^{(0)} = \widetilde{\mathbf{F}}\mathbf{y}. \tag{28}$$

**SCI**: Regarding the initialization, we derive $\mathbf{x}'^{(0)} \in \mathbb{R}^{H \times W \times N_\zeta}$ by repeating the measurement $\mathbf{y}$ $N_\zeta$ times along the channel dimension. Subsequently, the concatenation of $\mathbf{x}'^{(0)}$ and the 3D mask $\mathbf{A} \in \mathbb{R}^{H \times W \times N_\zeta}$ in channel dimension is inputted into a convolutional layer with a kernel size of $1 \times 1$, yielding the initial estimate $\mathbf{x}^{(0)} \in \mathbb{R}^{H \times W \times N_\zeta}$:

$$\mathbf{x}^{(0)} = \text{Conv}_1(\text{Concat}(\mathbf{x}'^{(0)}, \mathbf{A})). \tag{29}$$

### A.4 Implementation Details for Different CI Tasks

Table 5: The configurations of pretraining and fine-tuning on the ICS, CS-MRI and SCI tasks.

| Configuration | ICS | | CS-MRI | | SCI | |
|---|---|---|---|---|---|---|
| | Pretrain | Fine-tune | Pretrain | Fine-tune | Pretrain | Fine-tune |
| sampling matrix init | Gaussian matrix | - | Pseudo Radial mask | - | Pre-defined coded aperture | - |
| weight init | trunc.normal (0.2) | - | trunc.normal (0.2) | - | trunc.normal (0.2) | - |
| $\alpha_k$ init | 0.5 | - | 0.5 | - | 0.5 | - |
| $\lambda_k$ init | 1e-3 | - | 1e-3 | - | 1e-3 | - |
| block size | 32 | 32 | - | - | - | - |
| stages count | 5 | 5 | 5 | 5 | 5 | 5 |
| image size | 96×96 | 96×96 | 256×256 | 256×256 | 256×256 | 256×256 |
| basic channel count $C_0$ | 1 | 1 | 1 | 1 | 28 | 28 |
| stage channel count $[C_1, C_2, C_3]$ | [32, 48, 80] | [32, 48, 80] | [32, 48, 80] | [32, 48, 80] | [32, 48, 80] | [32, 48, 80] |
| number of heads $N_h$ | 8 | 8 | 8 | 8 | 8 | 8 |
| channel per head $C_h$ | 64 | 64 | 64 | 64 | 64 | 64 |
| batch size | 10 | 10 | 2 | 2 | 2 | 2 |
| training epochs | 400 | 400 | 200 | 100 | 400 | 400 |
| base learning rate | 2e-4 | 4e-5 | 1e-4 | 4e-5 | 1e-4 | 2e-5 |
| min learning rate | 1e-6 | 1e-6 | 1e-6 | 1e-6 | 1e-6 | 1e-6 |
| optimizer | AdamW (Loshchilov & Hutter (2019)) | AdamW | AdamW | AdamW | AdamW | AdamW |
| weight decay | 0.05 | 0.05 | 0.05 | 0.05 | 0.05 | 0.05 |
| optimizer momentum | 0.9, 0.999 | 0.9, 0.999 | 0.9, 0.999 | 0.9, 0.999 | 0.9, 0.999 | 0.9, 0.999 |
| warmup epochs | 5 | 5 | 5 | 5 | 5 | 5 |
| warmup schedule | linear | linear | linear | linear | linear | linear |
| learning rate schedule | cosine annealing | cosine annealing | cosine annealing | cosine annealing | cosine annealing | cosine annealing |
| time consumption | about 5 days | about 5 days | about 30 hours | about 15 hours | about 9 days | about 9 days |
| implementation | Pytorch 2.2.1 (Paszke et al. (2019)) | | | | | |
| CPU | 13th Gen Intel Core i9-13900KF | | | | | |
| GPU | RTX 4090 24 GB | | | | | |

**ICS**: For the ICS task, we employ a training dataset of 40,000 images randomly selected from the COCO2017 unlabeled image dataset (Lin et al. (2014)), with an additional 1,000 images reserved for validation. During training, we apply diverse data augmentation techniques, including random cropping, scaling, and rotation. Initially, the DUMoE model is trained with a sampling ratio of 0.25. Subsequently, fine-tuning is performed at various sampling ratios, leveraging the pretrained DUMoE weights from the initial training. Notably, the model jointly learns the sampling matrix. Furthermore, LPIPS scores are computed using VGG as the base network (Zhang et al. (2018)).

**CS-MRI**: In the CS-MRI experiments, our dataset consists of 100 training MRI images and 50 test MRI images sourced from the Brain dataset (Yang et al. (2020)) as used in previous works (Zhang & Ghanem (2018); Yang et al. (2020); Gan et al. (2024a)). All images share a uniform size of 256×256. During training, random rotation is applied as a data augmentation technique. The DUMoE model is first trained with a sampling ratio of 0.10, followed by fine-tuning at various sampling ratios using the pretrained weights.

**SCI**: The SCI experiments are conducted using both simulation and real HSI data. Following the settings of previous works (Ma et al. (2019); Meng et al. (2020); Huang et al. (2021); Hu et al. (2022); Cai et al. (2022b)), we select $N_\zeta = 28$ wavelengths ranging from 450 nm to 650 nm and $d = 2$ through spectral interpolation manipulation to derive HSIs. For simulations, the CAVE dataset (Park et al. (2007)), which contains thirty-two HSIs with a spatial size of $512 \times 512$, serves as the training set, while ten scenes from KAIST (Choi et al. (2017)) are utilized for testing. During training, data augmentation techniques such as random cropping into 256×256, slicing, and rotation are employed. In real data experiments, 11-bit shot noise is introduced into the measurements of CAVE and KAIST datasets during training to mimic real-world noise disturbances. Fine-tuning is performed based on the pretrained model using simulation data. Testing is conducted using five real scenes from the real CASSI system (Meng et al. (2020)).

Please refer to Tab. 5 for detailed configurations of DUMoE for the ICS, CS-MRI and SCI tasks.

Table 6: Average PSNR (dB) and SSIM performance comparisons of DUMoE and other ICS methods on various datasets at high sampling ratios (0.30, 0.40 and 0.50).

| Methods | Urban100 | | | | | | | | General100 | | | | | | | |
|---|---|---|---|---|---|---|---|---|---|---|---|---|---|---|---|---|
| | 0.30 | | 0.40 | | 0.50 | | Avg. | | 0.30 | | 0.40 | | 0.50 | | Avg. | |
| | PSNR | SSIM | PSNR | SSIM | PSNR | SSIM | PSNR | SSIM | PSNR | SSIM | PSNR | SSIM | PSNR | SSIM | PSNR | SSIM |
| CASNet (TIP 2022) | 33.35 | 0.9509 | 35.46 | 0.9668 | 37.46 | 0.9773 | 35.42 | 0.9650 | 39.32 | 0.9730 | 41.56 | 0.9827 | 43.74 | 0.9887 | 41.54 | 0.9815 |
| DGUNet+ (CVPR 2022) | 33.16 | 0.9510 | 35.24 | 0.9666 | 37.65 | 0.9785 | 35.35 | 0.9654 | 38.87 | 0.9724 | 41.07 | 0.9821 | 43.26 | 0.9884 | 41.07 | 0.9810 |
| FSOINet (ICASSP 2022) | 33.84 | 0.9540 | 35.93 | 0.9688 | 37.80 | 0.9777 | 35.86 | 0.9668 | 39.40 | 0.9735 | 41.62 | 0.9831 | 43.69 | 0.9887 | 41.57 | 0.9818 |
| TransCS (TIP 2022) | 32.01 | 0.9384 | 35.29 | 0.9649 | 37.28 | 0.9762 | 34.86 | 0.9598 | 37.81 | 0.9669 | 40.74 | 0.9806 | 42.89 | 0.9873 | 40.48 | 0.9782 |
| DPC-DUN (TIP 2023) | 33.53 | 0.9449 | 35.58 | 0.9622 | 37.52 | 0.9737 | 35.54 | 0.9603 | 37.76 | 0.9590 | 39.95 | 0.9731 | 42.00 | 0.9820 | 39.91 | 0.9713 |
| OCTUF (CVPR 2023) | 34.21 | 0.9555 | 36.25 | 0.9669 | 38.29 | 0.9797 | 36.25 | 0.9674 | 39.54 | 0.9740 | 41.76 | 0.9833 | 43.95 | 0.9892 | 41.75 | 0.9822 |
| NesTD-Net (TIP 2024) | 33.52 | 0.9516 | 35.96 | 0.9683 | 38.07 | 0.9786 | 35.85 | 0.9662 | 39.34 | 0.9732 | 41.78 | 0.9831 | 43.97 | 0.9891 | 41.70 | 0.9818 |
| LTwIST (TCSVT 2024) | 33.02 | 0.9477 | 35.16 | 0.9643 | 37.12 | 0.9753 | 35.10 | 0.9624 | 38.61 | 0.9699 | 40.85 | 0.9806 | 42.97 | 0.9872 | 40.81 | 0.9792 |
| UFC-Net (CVPR 2024) | 33.78 | 0.9524 | 35.93 | 0.9679 | 37.98 | 0.9782 | 35.90 | 0.9662 | 38.89 | 0.9704 | 41.17 | 0.9810 | 43.35 | 0.9878 | 41.14 | 0.9797 |
| **DUMoE (Our Method)** | **34.54** | **0.9576** | **36.58** | **0.9706** | **38.49** | **0.9797** | **36.54** | **0.9693** | **39.64** | **0.9743** | **41.96** | **0.9836** | **44.09** | **0.9894** | **41.90** | **0.9824** |

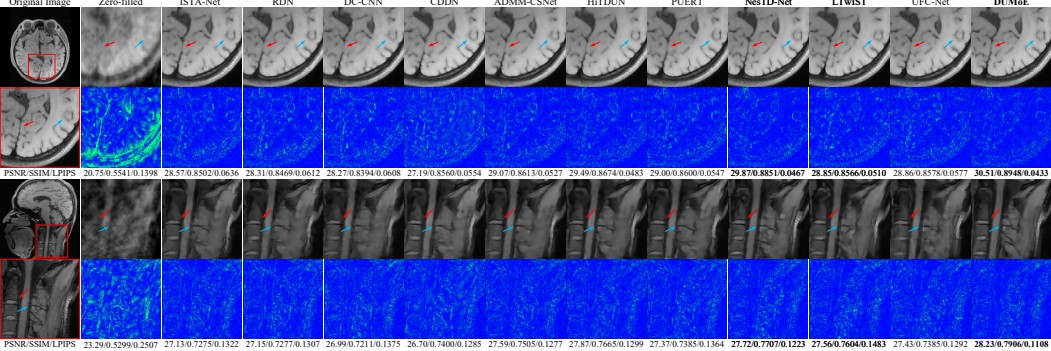

Figure 6: Comparisons of visual results with error maps and corresponding PSNR (dB)/SSIM/LPIPS performance between DUMoE and other advanced CS-MRI methods at a sampling ratio of 0.10.

## A.5 ADDITIONAL EXPERIMENTS

### A.5.1 ICS

We conduct qualitative comparisons between DUMoE and other ICS methods on Urban100 and General100 at high sampling ratios (0.30, 0.40, and 0.50). As shown in Tab. 6, our proposed DUMoE consistently outperforms other advanced methods at these high sampling ratios, with OCTUF and NesTD-Net achieving the second-best results. Specifically, on Urban100 at a sampling ratio of 0.30, DUMoE achieves PSNR improvements of approximately 1.01 dB (3.02%), 0.33 dB (0.96%), 1.52 dB (4.60%), 1.02 dB (3.04%), and 0.76 dB (2.25%) compared to DPC-DUN, OCTUF, LTwIST, NesTD-Net, and UFC-Net, respectively. Additionally, the SSIM improvements are approximately 0.0127 (1.34%), 0.0021 (0.22%), 0.0099 (1.04%), 0.0060 (0.63%), and 0.0052 (0.55%), respectively.

### A.5.2 CS-MRI

We present visual comparisons of reconstructed magnetic resonance (MR) images between DUMoE and other CS-MRI methods. As shown in Fig. 6, DUMoE exhibits superior performance in reconstructing fine details and enhancing human perception quality, with fewer errors compared to other methods in CS-MRI tasks.

### A.5.3 SCI

We present visual comparisons of reconstructed HSI between DUMoE and other SCI methods using both simulated and real HSI data. As illustrated in Fig. 7, the reconstructed HSI by DUMoE exhibits fewer artifacts and more accurate details compared to other SCI methods across various spectral channels. Additionally, the spectral density curves in the bottom left of Fig. 7, corresponding to the areas highlighted in the red boxes in the RGB image, demonstrate the highest correlation and alignment of DUMoE's spectral curves with the reference curves, highlighting the advantages of our proposed DUMoE in HSI reconstruction. Furthermore, Fig. 8 presents visual comparisons of DUMoE and other SCI methods on Scene 4 and Scene 5 using 2 spectral channels of real HSI data

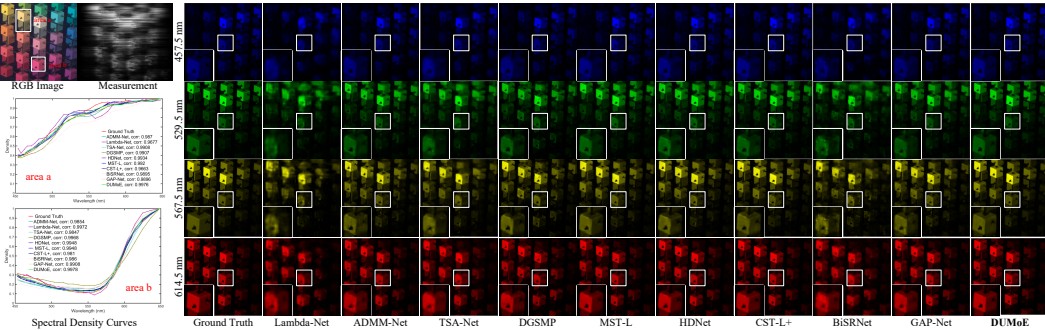

Figure 7: Simulation HSI reconstruction comparisons of DUMoE and other SCI methods on Scene 2 with 4 (out of 28) spectral channels.

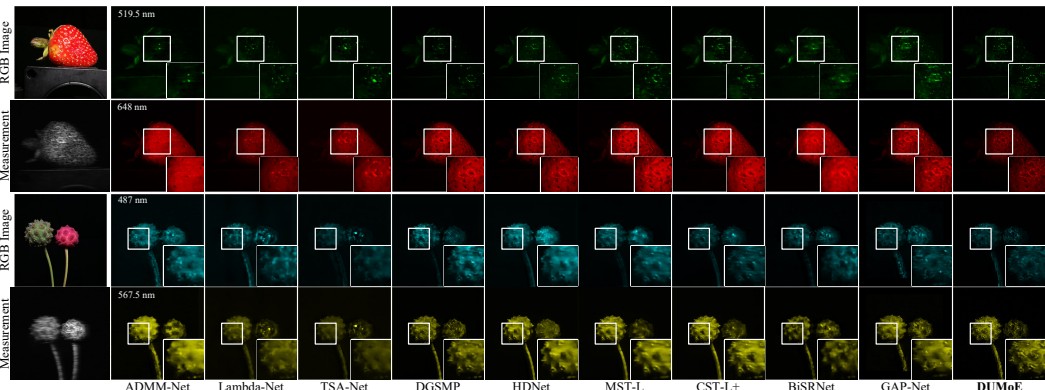

Figure 8: Real HSI reconstruction comparisons of DUMoE and other SCI methods on Scene 4 and Scene 5 with 2 (out of 28) spectral channels.

(Meng et al. (2020)), showcasing the superior performance of DUMoE on real HSI data. Moreover, Tab. 7 provides details on the number of parameters and FLOPs of different SCI methods on the KAIST dataset (Choi et al. (2017)).

Table 7: Number of parameters (M) and FLOPs (G) of different SCI methods on the KAIST dataset.

| Methods | ADMM-Net | Lambda-Net | TSA-Net | DGSMP | HDNet | MST-L | CST-L+ | RDFNet | GAP-Net | EDUNet | DWMT | DUMoE |
|---|---|---|---|---|---|---|---|---|---|---|---|---|
| Params. | 4.27 | 62.64 | 44.25 | 3.76 | 2.37 | 2.03 | 3.00 | 1.29 | 4.27 | 1.51 | 14.48 | 4.07 |
| FLOPs | 78.58 | 117.98 | 110.06 | 646.65 | 154.76 | 28.15 | 40.01 | 604.88 | 78.58 | 24.24 | 46.71 | 183.30 |

### A.5.4 VISUALIZATIONS OF DAM FOR VARIOUS CI TASKS

In this section, we present detailed visualizations of Degradation-Aware Mask (DAM) for three CI tasks. Specifically, we illustrate how DAM captures different types of degradation at the image-level domain $\mathbf{d}_1^{(k)}$, the measurement-level domain $\mathbf{d}_2^{(k)}$, and how the absolute sum of generated mask channels evolves across stages $k = 1, 3, 5$.

In ICS, as shown in Fig. 9, the image-level domain degradation $\mathbf{d}_1^{(k)}$ primarily reflects global image degradation and block artifacts, which are characteristic of the block sampling process in compressed sensing. Conversely, $\mathbf{d}_2^{(k)}$ is more focused on finer details such as edges and noise, which tend to be more vulnerable to degradation. As the number of stages increases, the mask progressively incorporates richer texture details.

For CS-MRI, as shown in Fig. 10, sampling is performed in the Fourier domain using a subsampling mask, resulting in aliasing artifacts. Here, both $\mathbf{d}_1^{(k)}$ and $\mathbf{d}_2^{(k)}$ capture different aspects of this

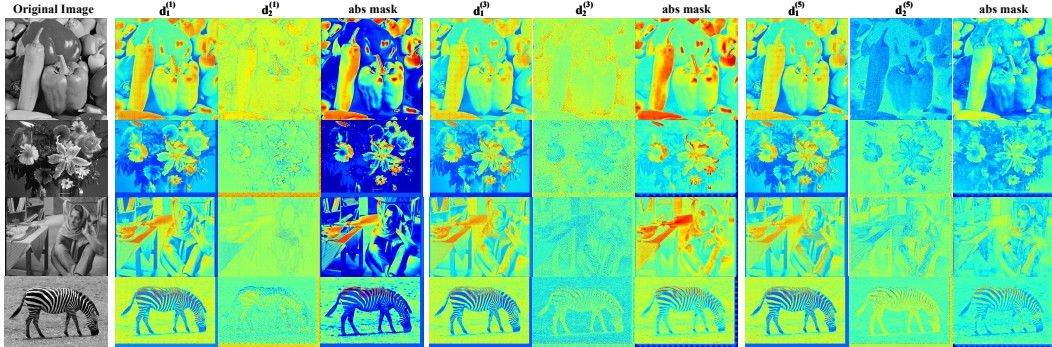

Figure 9: The visualizations of image-level domain degradation $\mathbf{d}_1^{(k)}$, measurement-level domain degradation $\mathbf{d}_2^{(k)}$ and absolute sum of generated mask channels at stages of $k = \{1, 3, 5\}$ for ICS at a sampling ratio of 0.25.

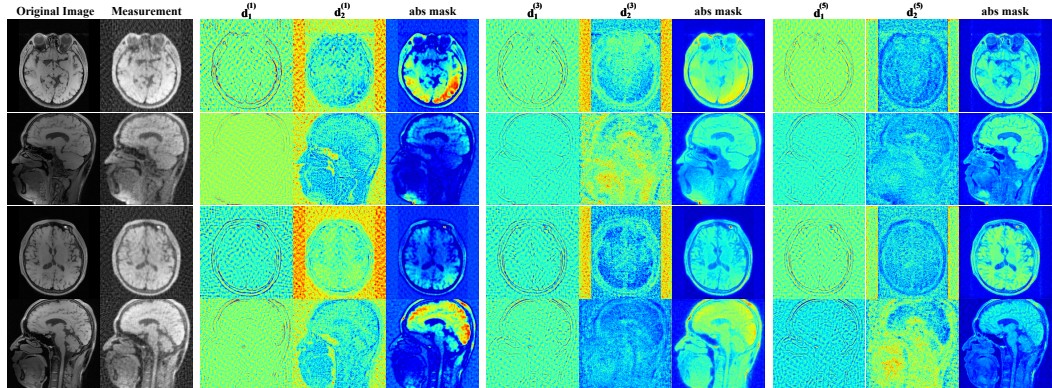

Figure 10: The visualizations of image-level domain degradation $\mathbf{d}_1^{(k)}$, measurement-level domain degradation $\mathbf{d}_2^{(k)}$ and absolute sum of generated mask channels at stages of $k = \{1, 3, 5\}$ for CS-MRI at a sampling ratio of 0.20.

degradation: $\mathbf{d}_1^{(k)}$ emphasizes edge information, while $\mathbf{d}_2^{(k)}$ focuses on broader degraded regions. The mask also becomes more refined with stage progression, revealing increasing detail.

The SCI task, as shown in Fig. 11, involves a 3D hyperspectral image compressed into a 1D measurement via a coded aperture. This process is prone to noise-induced artifacts. Initially, both

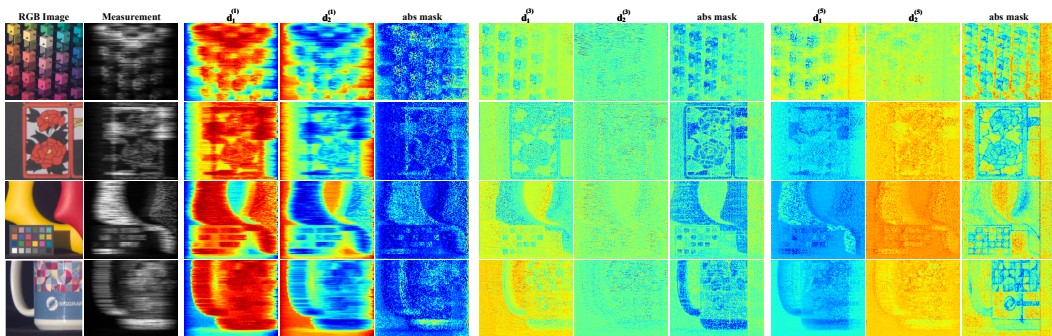

Figure 11: The visualizations of image-level domain degradation $\mathbf{d}_1^{(k)}$, measurement-level domain degradation $\mathbf{d}_2^{(k)}$ and absolute sum of generated mask channels at stages of $k = \{1, 3, 5\}$ for SCI.

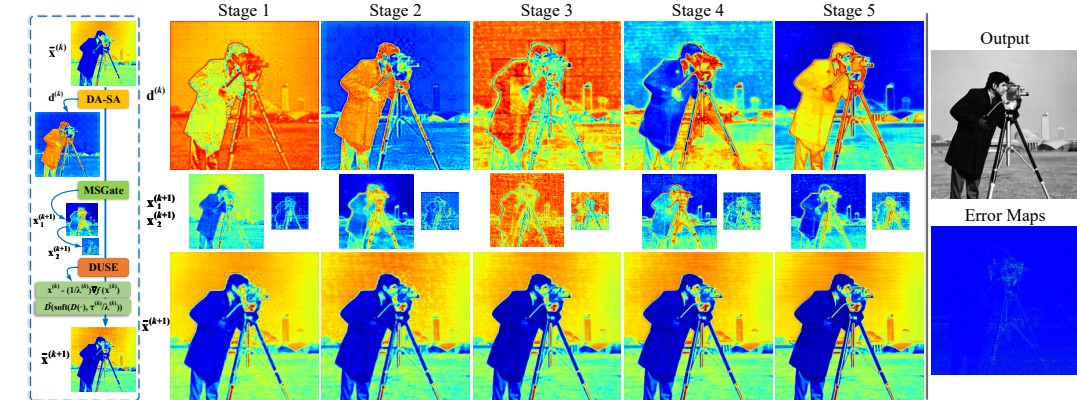

Figure 12: Feature visualization of the iteration stages in DUMoE at a sampling ratio of 0.10.

$\mathbf{d}_1^{(k)}$ and $\mathbf{d}_2^{(k)}$ capture these artifacts, but as the stages evolve, $\mathbf{d}_1^{(k)}$ emphasizes texture recovery, while $\mathbf{d}_2^{(k)}$ continues to highlight noise-affected areas. The mask becomes more precise in revealing important details as the stages progress.

As detailed in Fig. 5a, Tab. 4a and through the analysis in Sec. 5.1 of our paper, the DAM effectively guides DUMoE to focus on critical degraded image areas and fine details, despite variations in sampling, initialization processes, and data types (2D and 3D) across different CI tasks. This highlights the effectiveness and generalization of our proposed method, enhancing feature extraction capabilities across diverse CI tasks.

### A.5.5 VISUALIZATIONS OF IMAGE FEATURE MAPS

Fig. 12 visualizes the features across different iteration stages and modules in DUMoE, demonstrating the contributions and attention of different modules to the iterative image refinement during the reconstruction, thus enhancing the effectiveness of DUMoE.

Furthermore, as shown in Fig. 13, we present visualizations of image features and the corresponding top-1 selection of the DUSE for different images at each iteration stage across various sampling ratios. In the initial stages, DUMoE tends to capture the overall contour information of the images. However, at lower sampling ratios, block artifacts may be more prominent. Nevertheless, as the iterative stages progress, the details and texture information within the images become increasingly enriched, consequently diminishing block artifacts and resulting in high-fidelity image reconstruction. Notably, it is evident that at different sampling ratios, the refinement and enhancement of details and texture information in diverse images evolve through different experts during the iterative stages. This observation underscores the ability of DUMoE to dynamically select DUSE, facilitating iterative refinement tailored to the diverse characteristics of images during the iteration stages.

### A.5.6 ABLATION STUDIES ON DUSE NUMBER

Table 8: The performance comparisons of cases under different number of DUSE.

| DUSE | 1 | 2 | 3 | 5 | 3 |
|---|---|---|---|---|---|
| Switch Routing | - | w/ | w/ | w/ | w/o |
| PSNR (dB) | 34.31 | 34.39 | 34.42 | 34.38 | 34.39 |
| SSIM | 0.9325 | 0.9332 | 0.9334 | 0.9332 | 0.9329 |
| Params. (M) | 3.99 | 4.08 | 4.17 | 4.34 | 4.17 |
| FLOPs (G) | 142.34 | 142.34 | 142.34 | 142.34 | 158.72 |

We conduct an ablation study on the number of experts as shown in Tab. 8. As the number of DUSE increases, performance gradually improves, peaking at three blocks. However, performance declines

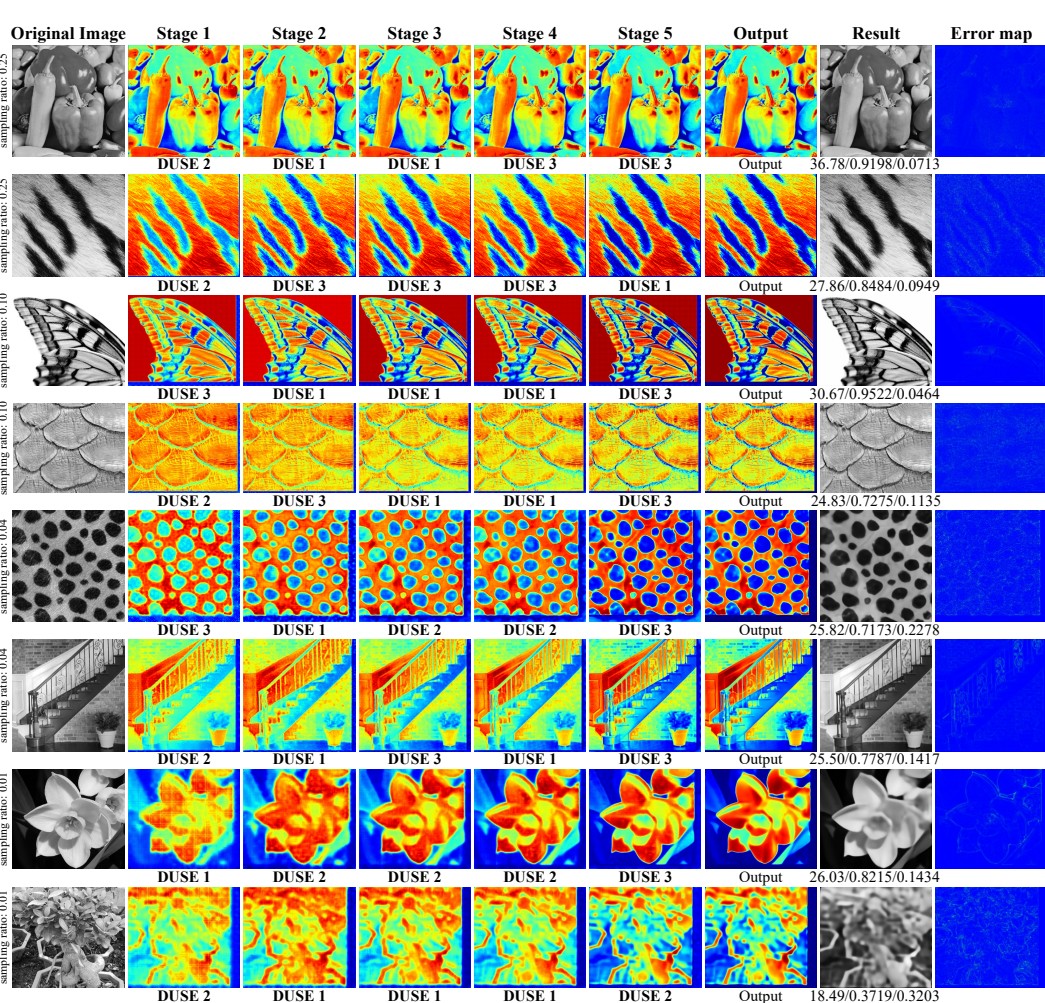

Figure 13: Visualizations of image features and the corresponding top-1 selection of the DUSE for diverse images in each iteration stage at different sampling ratios.

when the number of DUSE reaches five, likely due to increased training complexity and the higher number of parameters, requiring more epochs to achieve optimal performance.

## A.6 LIMITATIONS

In our main paper, we primarily focus on utilizing the $\ell_1$-norm as the image prior to enforce sparsity in the transform domain and employ soft thresholding to address Eq. (8). However, it's worth noting that various other image priors exist, including the $\ell_0$-norm, total variation, low-rank, etc., for different applications. Besides, many deep unfolding-based methods leverage deep denoising networks to replace image prior terms (Song et al. (2023c); Mou et al. (2022); Cai et al. (2022c)). Moving forward, we aim to explore a broader spectrum of image prior terms, thereby enhancing the versatility of DUMoE to address a wider array of image ill-posed problems, such as image super-resolution, image deraining, image denoising, etc.

Furthermore, the proposed DUMoE framework is designed as a general approach capable of addressing a broad range of CI tasks, including ICS, CS-MRI, and SCI, without being restricted to a single domain. This generality allows DUMoE to be applied to various CI tasks with diverse data characteristics (e.g., 2D and 3D) and requirements (e.g., different sampling and initialization processes). However, this broad applicability may result in performance trade-offs for highly specialized tasks. We view this as an area for future improvement, where incorporating more specialized knowledge into the DUMoE framework could potentially mitigate these trade-offs and yield more competitive results in specialized applications.

## A.7 CODE SUBMISSION AND REPRODUCIBILITY

We submit the source code and pre-trained models in the supplemental material and provide the detailed experimental settings for reproducing the results presented in our paper. Additionally, both the source code and pre-trained models will be publicly released for broader accessibility and reproducibility.

