# OpenReview forum: "DUMoE: Deep Unfolding Mixture-of-Experts for Compressive Imaging"
_ICLR.cc/2025/Conference — ICLR 2025 Conference Withdrawn Submission_

### Official Review · Reviewer_gjLQ · 2024-10-24

**Soundness:** 2
**Presentation:** 2
**Contribution:** 1
**Rating:** 5
**Confidence:** 5

**Summary:**

This paper presents DUMoE, a sparsely-activated Deep Unfolding Mixture-of-Experts (MoE) architecture for Compressive Imaging, which leverages adaptive expert selection and introduces mechanisms like top-1 switch routing, a Degradation-Aware Mask, and a Multi-Scale Gate to enhance reconstruction performance and efficiency.

**Strengths:**

Introducing MoE to deep unfolding  networks.

**Weaknesses:**

The motivation and problem this paper aims to solve are not clearly articulated. In the abstract, the authors state that "existing DUNs often lack flexibility in handling details and features in different images." While the reference to handling "details" makes sense, the term "handling features" is vague and needs clarification. Additionally, the authors should provide evidence for the claim made in the abstract that "existing DUNs often lack flexibility in handling details and features in different images during reconstruction." The iteration process in DUNs is driven by an optimization framework derived from the degradation equation of specific imaging systems, so the claim of inflexibility requires stronger justification. Moreover, how the proposed method solves this issue, given that it also follows a deep unfolding framework with coarse-to-fine iterations, remains unclear.

In lines 66-67, the authors claim that "existing DUNs often employ multiple iterative modules cascading through the same module for each iteration, limiting flexibility when handling fine-grained details in diverse images." However, if the parameters of the denoiser in each stage are not shared and each denoiser has its own parameters, this flexibility issue might not exist. It would be helpful if the authors could provide further analysis on this point.

**Questions:**

The experimental results do not convincingly demonstrate the effectiveness of the proposed approach. The introduction of the MoE with a complex gating mechanism shows only marginal improvements, as reflected in Table 4-(c). For instance, DGUNet achieves 20.15dB PSNR, while the proposed model achieves 20.33dB PSNR on the Urban100 dataset, with a significant increase in inference time (DGUNet: 27±1 vs the proposed model: 78±6). Additionally, the visual comparisons in Figure 3 do not show clear improvements, making it difficult to assess the claimed visual enhancements. The authors should consider providing more persuasive evidence and analysis to support their contributions.

Is it possible to make the proposed model lightweight via simplifying the gate module?

---

### Official Review · Reviewer_MC6U · 2024-10-29

**Soundness:** 2
**Presentation:** 3
**Contribution:** 2
**Rating:** 3
**Confidence:** 4

**Summary:**

This paper aims to incorporate the idea of the Mixture of Experts into the deep unfolding network for compressed sensing.

**Strengths:**

Comprehensive comparisons with recent methods.
Good visual results.

**Weaknesses:**

- The methods mainly involve adding gating mechanisms within UNet and replacing fixed sparsifying transform with adaptive network layers. None of these are new contributions.

- The advantages of combining MoE with DUNs have not been sufficiently demonstrated. The claim that MoE helps DUNs flexibly handle features of different images is not backed up by any experiment/visual analysis.

- The ablation study should include the baseline results where none of the proposed components were present. Currently, it seems that each component's improvement is incremental.

**Questions:**

See above.

---

### Official Review · Reviewer_wSKv · 2024-11-01

**Soundness:** 4
**Presentation:** 4
**Contribution:** 4
**Rating:** 8
**Confidence:** 4

**Summary:**

This paper proposes adapting the MoE (Mixture of Experts) architecture for compressive imaging. The authors introduce several novel ideas to map the original MoE design into this domain. Specifically, they replace conventional self-attention with degradation-aware self-attention (DA-SA), which incorporates measurement and image errors as inputs. They substitute the standard sparse gating mechanism with a multi-scale gate (MSGate) and replace the feed-forward network (FFN) with DUSE, a one-step computational update within the network. The proposed model is validated on various tasks, including natural image compressive sensing, magnetic resonance imaging, and snapshot compressive imaging, demonstrating consistent performance improvements. In general, I admire the interesting ideas and extensive efforts for experiments and code release.

**Strengths:**

1. The paper is well-written with convincing results, making it accessible even to non-experts.
2. The motivations behind the DA-SA, MSGate, and DUSE designs are clearly articulated.
3. The study includes extensive evaluations across various image compression tasks, providing a thorough analysis.

**Weaknesses:**

**Major Limitations:**

1. **Redundant Inputs in DA-SA:** The inputs for image and measurement errors in DA-SA are redundant, as they can be derived from one another through computational inversion. Please provide a justification for this design choice, ideally supported by an ablation study.

2. **Application of Degradation Mask in DA-SA:** The study states, “the obtained features are combined with Value \(V\) in DA-SA using a Hadamard product to prioritize attention to the degraded parts.” Why is the degradation mask applied to the value rather than the attention map (\(Q \times K\)), as is common with conventional attention masking? Could you clarify if there is a specific reasoning or detail I missed?

3. **MSGate Convolution Design:** The 2 × 2 convolution with a stride of 2 in MSGate may introduce checkerboard effects, as it is not a typical design compared to 3 × 3 convolution with overlap. Consider clarifying this choice or addressing potential implications.

4. **Additional Metrics Beyond PSNR and SSIM:** While PSNR and SSIM are established metrics, they have limitations (e.g., PSNR’s preference for over-smoothed images). Adding perceptual-, distribution-, or user-based metrics could enhance the evaluation.

5. **Clarification on Figure 4 Visualizations:** Could you specify which pretrained model was used in the visualizations? Additionally, in Figure 4a, DUSE2 appears absent; was this branch not activated? In Figure 4b, the distinct hidden states across experts are remarkable and seem complementary. This is impressive but could benefit from further elaboration.

6. **Ablation Study on DUSE Number:** In section A.5.6, does increasing the training dataset mitigate the performance degradation observed with a higher DUSE number?

7. **Training Stability with MoE:** Training MoE can be challenging due to sparse routing dynamics, especially on smaller datasets. Could you report training dynamics to provide insights on stability?

---

**Minor Limitations:**
1. **Claim of Novelty in MoE for Compressive Imaging:** The claim “first MoE for compressive imaging” could benefit from clarification, as similar ideas have been proposed, such as in Nguyen et al. (2023), *Bayesian Nonparametric Mixture of Experts for Inverse Problems*. Please clarify how this work differentiates from such studies.

2. **Code Documentation:** The submitted code would be more accessible with example data and expected outputs in the documentation. Currently, only the sampling matrix and model weights are provided, which makes it challenging to establish the evaluation pipeline.

**Questions:**

Please see my comments in Weakness.

---

### Note · Authors · 2024-11-14

I have read and agree with the venue's withdrawal policy on behalf of myself and my co-authors.